# Cryogenic contrast-enhanced microCT enables nondestructive 3D quantitative histopathology of soft biological tissues

Arne Maes ®[1,2,3], Camille Pestiaux ®[2,3], Alice Marino[4], Tim Balcaen ®[2,3,5], Lisa Leyssens ®[2,3], Sarah Vangrunderbeeck ®[2,3,5], Grzegorz Pyka[2,3], Wim M. De Borggraeve ®[5], Luc Bertrand ®[4], Christophe Beauloye[6], Sandrine Horman ®[4], Martine Wevers[1] & Greet Kerckhofs ®[1,2,3,7] ✉

Biological tissues comprise a spatially complex structure, composition and organization at the microscale, named the microstructure. Given the close structure-function relationships in tissues, structural characterization is essential to fully understand the functioning of healthy and pathological tissues, as well as the impact of possible treatments. Here, we present a nondestructive imaging approach to perform quantitative 3D histo(patho)logy of biological tissues, termed Cryogenic Contrast-Enhanced MicroCT (cryo-CECT). By combining sample staining, using an X-ray contrast-enhancing staining agent, with freezing the sample at the optimal freezing rate, cryo-CECT enables 3D visualization and structural analysis of individual tissue constituents, such as muscle and collagen fibers. We applied cryo-CECT on murine hearts subjected to pressure overload following transverse aortic constriction surgery. Cryo-CECT allowed to analyze, in an unprecedented manner, the orientation and diameter of the individual muscle fibers in the entire heart, as well as the 3D localization of fibrotic regions within the myocardial layers. We foresee further applications of cryo-CECT in the optimization of tissue/food preservation and donor banking, showing that cryo-CECT also has clinical and industrial potential.

Biological tissues comprise a spatially complex and heterogeneous structure and organization at the microscale, also called the tissue microstructure. Given the close structure-function relationship in biological tissues, the ability to visualize and quantitatively investigate the tissue microstructure in 3D is crucial. It is even more important for diseased tissues as pathologies are regularly associated with structural alterations leading to tissue and organ dysfunction[1]. Currently, the gold standard for ex vivo tissue imaging remains classical 2D histological assessment thanks to its high discriminative power, the wide range of available (counter)stains and the ability of performing immunohistochemistry or fluorescence microscopy. In classical 2D histology, the dissected tissue is embedded, sectioned and investigated using optical or electron microscopy[2]. Prior to microscopy, the sections are generally stained to highlight, for example, specific cells or various constituents of the extracellular matrix (ECM)[3]. Despite its many advantages, classical 2D histology is inherently limited by its 2D

[1]Department of Materials Engineering, KU Leuven, Heverlee, Belgium. [2]Biomechanics lab, Institute of Mechanics, Materials and Civil Engineering, UCLouvain, Louvain-la-Neuve, Belgium. [3]Pole of Morphology, Institute of Experimental and Clinical Research, UCLouvain, Brussels, Belgium. [4]Pole of Cardiovascular Research, Institute of Experimental and Clinical Research, UCLouvain, Brussels, Belgium. [5]Molecular Design and Synthesis, Department of Chemistry, KU Leuven, Leuven, Belgium. [6]Division of Cardiology, University Hospital Saint-Luc, Brussels, Belgium. [7]Prometheus, Division for Skeletal Tissue Engineering, KU Leuven, Leuven, Belgium. ✉e-mail: greet.kerckhofs@uclouvain.be

nature and the single sectioning orientation[4]. The intricate 3D tissue microstructure is, therefore, only partially revealed using classical 2D histology. In addition, this technique can be prone to image artefacts, such as sample distortion, folds, cracks and shrinkage due to dehydration. Information about the third dimension can be obtained to some extent by either serial stacking of 2D sections[5,6], or by applying more advanced optical imaging methods such as confocal microscopy[7,8], light sheet microscopy[9,10] or optical coherence tomography[11,12]. However, these imaging techniques are limited either by the intensive manual interaction and the limited resolution in the stacking direction (for serial stacking)[13], or by the limited penetration depth of light, often requiring sample transparency[14].

Microfocus X-ray computed tomography (microCT) offers a valuable solution for X-ray based 3D histology of biological tissues, complementary to classical 2D histology. The digital nature of microCT allows qualitative and quantitative 3D microstructural analysis of tissues and of their constituents. Moreover, the virtual slicing is not restricted to a single orientation and avoids sample destruction. MicroCT has been widely used to image mineralized tissues thanks to the large difference in X-ray attenuation of these tissues compared to the surrounding soft tissues. In order to extend microCT to the visualization of soft tissues, the sample can be stained prior to imaging using contrast-enhancing staining agents (CESAs). MicroCT combined with the use of CESAs is referred to as contrast-enhanced microCT (CECT) (reviewed in[15-18]), and has been used for X-ray based 3D histology of various types of soft tissues and organs including muscle[19-24], cartilage[25-30], ligaments[31,32], tendons[31-34], nervous tissue[23,35], kidneys[36,37], placenta[38], cardiovascular tissues[39-41], and the bone marrow compartment[36]. It is worth mentioning that phase-contrast microCT offers an alternative method to enhance soft tissue contrast[42,43]. However, this technique requires either the use of a dedicated synchrotron light source[44] or a highly specialized laboratory-based system with required scan times often exceeding several hours to days[33,45]. Moreover, the sample preparation can be tedious and frequently includes dehydration of the sample, potentially altering the tissue microstructure[33]. Micro-magnetic resonance imaging (micro-MRI) has also recently emerged as a promising 3D histological imaging modality thanks to its inherently high soft tissue contrast[46,47]. However, compared with microCT, the spatial resolution of micro-MRI is relatively low, with voxel sizes ranging only from tens to hundreds of microns[14].

Since the first reports on CECT about 15 years ago[48,49], several CESAs have been developed and established[50]. Among the most well-known CESAs are inorganic iodine solutions such as Lugol's iodine ($I_2KI$), Osmium tetroxide ($OsO_4$), Ioxaglate (Hexabrix), CA4+, and certain polyoxometalates (POMs) such as Phosphotungstic acid (PTA) and Phosphomolybdic acid (PMA)[15,51]. Despite the recent breakthroughs in the field of CECT, the nondestructive visualization of several soft tissues and organs, without inducing tissue shrinkage and deformation, remains challenging[31,52]. For instance, iodine-based CESAs, such as Lugol's iodine solution, have been reported to induce substantial tissue shrinkage with relative muscle volume shrinkage ranging from 25 to 65% depending on the CESA concentration[53]. Similarly, tendon fibers have been visualized with protocols involving inorganic iodine staining solutions[31,32] or storage media containing ethanol[34], which are both methods known to induce dehydration and tissue shrinkage. More recently, our research group introduced POM-based CESAs, which enhance soft tissue contrast while avoiding tissue shrinkage and deformation[36,38,54]. However, at the currently attainable spatial resolution of lab-based microCT, CECT imaging with these CESAs still does not allow to visualize fine microstructural details such as individual muscle or collagen fibers.

Here, we present an extended approach to CECT, termed cryogenic CECT or cryo-CECT, which enables nondestructive 3D histology of various individual soft tissue constituents by imaging the stained sample in its frozen state. While the staining with a CESA provides an overall soft tissue contrast, the freezing reveals the individual soft tissue constituents thanks to segregation between the CESA and the water contained in the tissue upon ice crystallization. Using cryo-CECT, we were able to nondestructively visualize in 3D individual skeletal muscle and tendon fibers, which could not be achieved with conventional CECT. Subsequently, we showed the added value of CECT and cryo-CECT compared to classical 2D histology for the microstructural characterization of the murine myocardium and to better describe the microstructural changes caused by transverse aortic constriction (TAC) surgery, a commonly used experimental model for pressure overload-induced cardiac hypertrophy.

## Results

### Cryo-(CE)CT enables nondestructive 3D visualization of individual skeletal muscle fibers

Muscle tissue has been the tissue of interest for many CECT research studies[19-24], in which individual muscle fibers or fascicles are often visualized by staining with inorganic iodine solutions, such as Lugol's iodine. However, substantial tissue shrinkage has been reported and is presumably the cause of visualization of individual fibers. Here, we demonstrate the nondestructive visualization of individual muscle fibers using cryo-(CE)CT. Two CESAs were compared for their use in cryo-CECT: Lugol's iodine ($I_2KI$) and 1:2 hafnium(IV)-substituted Wells-Dawson polyoxometalate (Hf-WD POM; $K_{16}[Hf(\alpha_2\text{-}P_2W_{17}O_{61})_2]\cdot19H_2O$). Samples immersed in phosphate-buffered saline (PBS, 1x) were used as control samples.

First, we quantified the degree of relative muscle tissue shrinkage/swelling, compared to the fixed state, for both CESAs during the staining process (Supplementary Fig. 1). The fixation resulted in an average relative volume shrinkage of about 8% compared to the fresh state. Despite the isotonicity of the staining solution, the Lugol's iodine staining caused a rapid initial tissue shrinkage (−19% after 1 day), which continued to increase during the staining process (−37% after 29 days). This tissue shrinkage was accompanied by a rapid decrease in the pH of the staining solution. In contrast to Lugol's iodine, Hf-WD POM staining resulted in a slight tissue swelling (+7% after 7 days). Interestingly, although complete staining was achieved after 7 days, the relative volume expansion continued to increase up to 18% after 27 days. Statistical analysis showed that the relative volume difference, compared to the fresh state, was already highly significant from the first day of staining with Lugol's iodine, whereas, for Hf-WD POM staining, the relative volume difference reached statistical significance only after 11 days of staining (Supplementary Fig. 1). Incubation in PBS did not cause significant changes in volume.

Imaging the unstained tissue at room temperature did not reveal the tissue microstructure, with the exception of a slight gray value difference between muscle tissue and adipose tissue (Fig. 1a). Although conventional CECT at room temperature using Lugol's iodine or Hf-WD POM enhanced this image contrast, it failed to effectively visualize the individual muscle fibers within the fascicles (Fig. 1b–c). For both CESAs, staining allowed the visualization of muscle fascicles, the intermuscular adipose tissue, blood vessels and the perimysium. Fast freezing the unstained tissue, by submerging it in isopentane at −78 °C, and imaging it in the frozen state revealed the individual muscle fibers within the fascicles, albeit at low image contrast (Fig. 1d). Performing this freezing step on samples that have been stained with Hf-WD POM or Lugol's iodine (i.e., cryo-CECT) substantially enhanced the visualization of the muscle fibers (Fig. 1e, f). Registering the cryo-(CE)CT images to the matching 2D histological section validated the visualization of the muscle (sub-)fascicles and muscle fibers (Fig. 1g–i). Compared to Lugol's iodine staining (Fig. 1e), individual muscle fibers were easier to distinguish using Hf-WD POM staining (Fig. 1f). For this reason, we further focused on Hf-WD POM as the CESA of choice for cryo-CECT imaging.

To perform cryo-(CE)CT, a dedicated in-situ microCT cryo-stage was developed in-house to maintain samples in the frozen state during

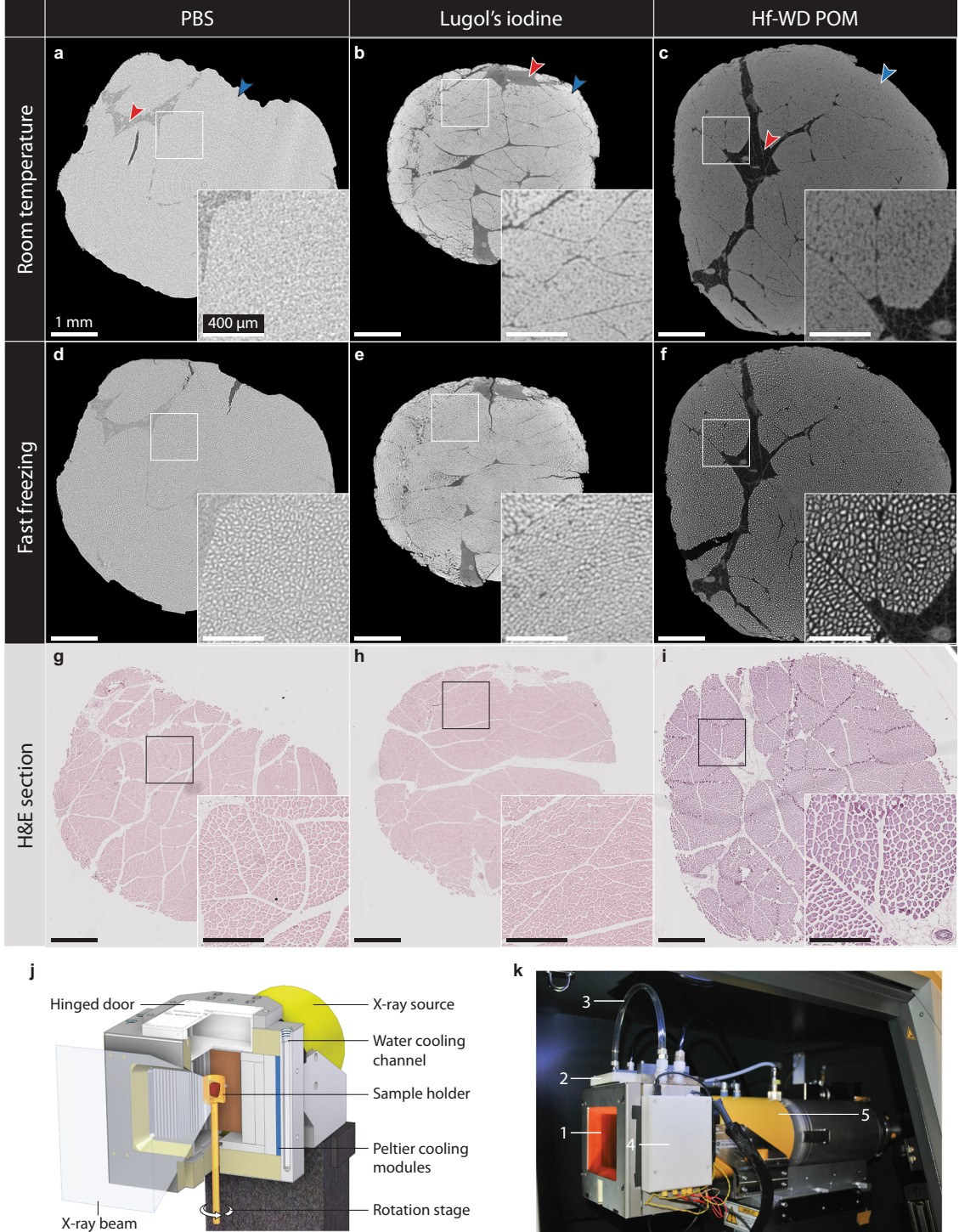

**Fig. 1 | The effect of freezing and different CESAs on the microstructure of bovine muscle tissue as visualized by CECT and cryo-(CE)CT.** **a**–**c** Conventional (CE)CT images of bovine muscle tissue acquired at room temperature without staining (**a**), stained with Lugol's iodine (**b**) and stained with Hf-WD POM (**c**). The muscle tissue (blue arrow) and the intermuscular fat (red arrow) can be distinguished. Histograms were windowed based on their dynamic range. Hence, gray values were not normalized in-between different datasets. **d**–**f** Cryo-(CE)CT images after freezing the tissues by submersion in isopentane at −78 °C (fast freezing). Individual muscle fibers (bright) surrounded by the fascicular matrix (dark) are visualized (*n* = 3 for each condition). **g**–**i** Classical 2D histological sections (H&E staining) matching the microCT images (*n* = 3 for each condition). Scale bars of the overview image and inset image correspond to 1 mm and 400 μm, respectively. **j**, **k** The in-house developed in-situ microCT cryo-stage. **j** 3D rendering of the computer-aided design with a cut-out, showing the individual components of the in-situ cryo-stage. **k** Photograph of the in-situ cryo-stage installed in the microCT system showing the sample chamber closed off by a radiolucent polyimide film (1), the hinged door (2), the tubes for liquid cooling (3), the enclosed circuit board (4) and the X-ray source (5).

CT scans while imaging at the maximum attainable spatial resolution (Fig. 1j, k) This was achieved by designing the stage in such a way that the minimum distance between the sample and the focal point of the X-ray source was not restricted by the stage. Commercial in-situ cooling stages based on contact cooling are often relatively bulky, which increases the minimum distance between the sample and the X-ray source, hence reducing the attainable spatial resolution. Furthermore, our cooling stage does not rely on contact cooling but, instead, comprises a chamber of air that is cooled down to −35 °C by six Peltier modules. The homogenous air cooling achieved by our cryo-stage avoids undesirable temperature gradients that are often associated with contact cooling. It is worth highlighting that the cryo-stage was not used to freeze samples, but solely to keep the samples frozen during scanning.

### The freezing rate is an important factor in the accurate visualization of the tissue microstructure

As shown in the previous section, selecting a suitable CESA for cryo-CECT imaging is crucial to obtain optimal visualization of the muscle fibers while preserving the tissue microstructure. Here, we investigated the effect of the freezing rate on both the visualization of muscle fibers and the tissue integrity. Qualitatively, slow freezing in air at −80 °C (SF) resulted in a larger spacing between the muscle fibers and fascicles in comparison with fast freezing by submersion in isopentane at −78 °C (FF) (Supplementary Fig. 2). This implies that SF causes a compression of the muscle fibers and an expansion of the interfascicular spaces. Occasionally, slow-frozen fibers were also found to coalesce and form clusters. By imaging the same sample after FF and after SF, we obtained a 3D fiber model of the individual muscle fibers inside a consistently located volume-of-interest (VOI) (Fig. 2a). Quantitative analysis of this fiber model showed a systematic and significant reduction in fiber diameter for SF, compared to FF (Fig. 2b). This confirms the qualitative finding that SF compresses the muscle fibers and, therefore, substantially alters the tissue microstructure. Concerning the fast freezing protocol, altering the temperature of the isopentane to −20 °C or −160 °C did not significantly affect the median fiber diameter (Supplementary Fig. 2). Although SF resulted in a compression of the muscle fibers, the tortuosity and the orientation of the fibers were not significantly altered compared to FF (Fig. 2c, d). This implies that, although the fiber is compressed radially, the fiber orientation is conserved after slow freezing. In combination with our observation that Hf-WD POM staining does not induce tissue shrinkage and for practical purposes concerning the freezing process, we propose that Hf-WD POM staining combined with fast freezing by submersion in isopentane at −78 °C is the preferred cryo-CECT protocol for muscle tissue.

Similar as for the muscle tissue, different freezing methods were tested to optimize the visualization of tendon's collagen fibers using cryo-CECT, while preserving the integrity of the tissue. For tendon tissue, as opposed to muscle tissue, increasing the temperature of the isopentane from −78 °C to −20 °C resulted in an enhanced image contrast between the collagen fibers and the surrounding matrix (Fig. 3a), which aided the segmentation of the fibers in postprocessing. Slow freezing in air at −80 °C resulted in large spaces surrounding the collagen fibers indicating excessive fiber compression. Therefore, submersion in isopentane at −20 °C is proposed as the optimal freezing method for the visualization of tendon collagen fibers, which was validated by classical 2D histology (Fig. 3b). The difference between the optimal freezing method for tendon tissue and muscle tissue is presumably linked to the dissimilar constitution of their fibers: collagen I fibrils and myofibrils, respectively, which could respond differently to the freezing process. This demonstrates the need for tissue-specific optimization of the freezing rate. The volume renderings of a few segmented collagen fibers show the unidirectional and longitudinal orientation of the fibers, typical for tendon tissue (Fig. 3c)[55]. As opposed to the collagen fibers in the tendon tissue (white inset in Fig. 3a), cryo-CECT did not allow the visualization of the individual collagen fibers within the non-mineralized fibrocartilaginous layer of the Achilles bone-tendon interface (star in Fig. 3a). However, the transition from tendon to non-mineralized fibrocartilage, in which the tendon fibers splay out in thinner interface fibers[34], could be more readily recognized in the cryo-CECT images compared to conventional CECT.

### The fast-frozen microstructure is stable during long-term storage at −80 °C and −20 °C

We showed that fast freezing conserves the fibrous microstructure of muscle tissue better than slow freezing. However, ice recrystallization might occur during long-time storage of the sample, leading to potential changes of the frozen tissue structure over time. Ensuring that the fast-frozen microstructure does not gradually evolve towards the thermodynamically more stable slow-frozen microstructure is important both for sample storage purposes at low temperatures and for cryo-CECT acquisitions, performed at an ambient temperature of −35 °C, with long acquisition times. To evaluate the stability of the fast-frozen fibrous microstructure, we imaged the muscle tissue using cryo-CECT immediately after fast freezing and again after storage for either 1 or 4 weeks. The 4-week samples were imaged again after storage for 23 months (100 weeks) to evaluate the long-term stability. This was done for storage both at −80 °C and at −20 °C. Quantitative 3D analysis of the fiber diameter within a consistently located VOI showed no significant differences in median fiber diameter after 1, 4 or 100 weeks of storage at either −80 °C or −20 °C (Fig. 4). The frozen tissue's microstructure is thus stable for at least 23 months at both storage temperatures. This finding considerably increases the experimental flexibility by allowing a considerable time buffer between freezing and imaging, and moreover, ensures no microstructural changes during imaging.

### 3D histopathology using cryo-CECT of pressure overload-induced hypertrophic murine hearts allows quantitative analysis of individual cardiac muscle fibers

To show the added value of cryo-CECT for 3D histopathology, compared to classical 2D histological assessment, we applied cryo-CECT to murine hearts subjected to pressure overload following TAC surgery (Fig. 5). The TAC surgery is a well-established technique to induce left ventricular concentric hypertrophy due to a chronic pressure overload. In our study, this pathological condition was reflected by an increase in both the heart to body weight ratio and the ratio of heart volume to body weight of the TAC group, compared to the sham-operated control group (Fig. 5a, b). The success of the TAC surgery was demonstrated by the increased aortic peak flow velocity measured at the site of constriction by Doppler echocardiography (Supplementary Table 1).

Conventional Hf-WD POM-based CECT allowed the visualization of the myocardium in its entirety, along with the heart valves, the main vessels of the heart and the papillary muscles with the chordae tendineae (Fig. 5c). However, Individual cardiac muscle fibers were only revealed using cryo-CECT (fast freezing in isopentane at −78 °C) (Fig. 5d, e). Moreover, the application of cryo-CECT allowed to perform full 3D orientation analysis of the individual cardiac muscle fibers (Fig. 5f–h). Using the polar angle θ, or inclination, of the cardiac muscle fibers in relation to the long axis of the interventricular septum, a distinction could be made between vertical fibers (θ ~ 0°), found at the subendocardial (inner) and subepicardial (outer) layers of the myocardium, and circumferential fibers (θ ~ 90°), located in the midmyocardium. For both animal groups, the typical helical winding of the superficial myocardium layers around the ventricles could be observed at the outer surface of the heart (Supplementary Video 1). Our results demonstrated that despite the hypertrophy, the overall

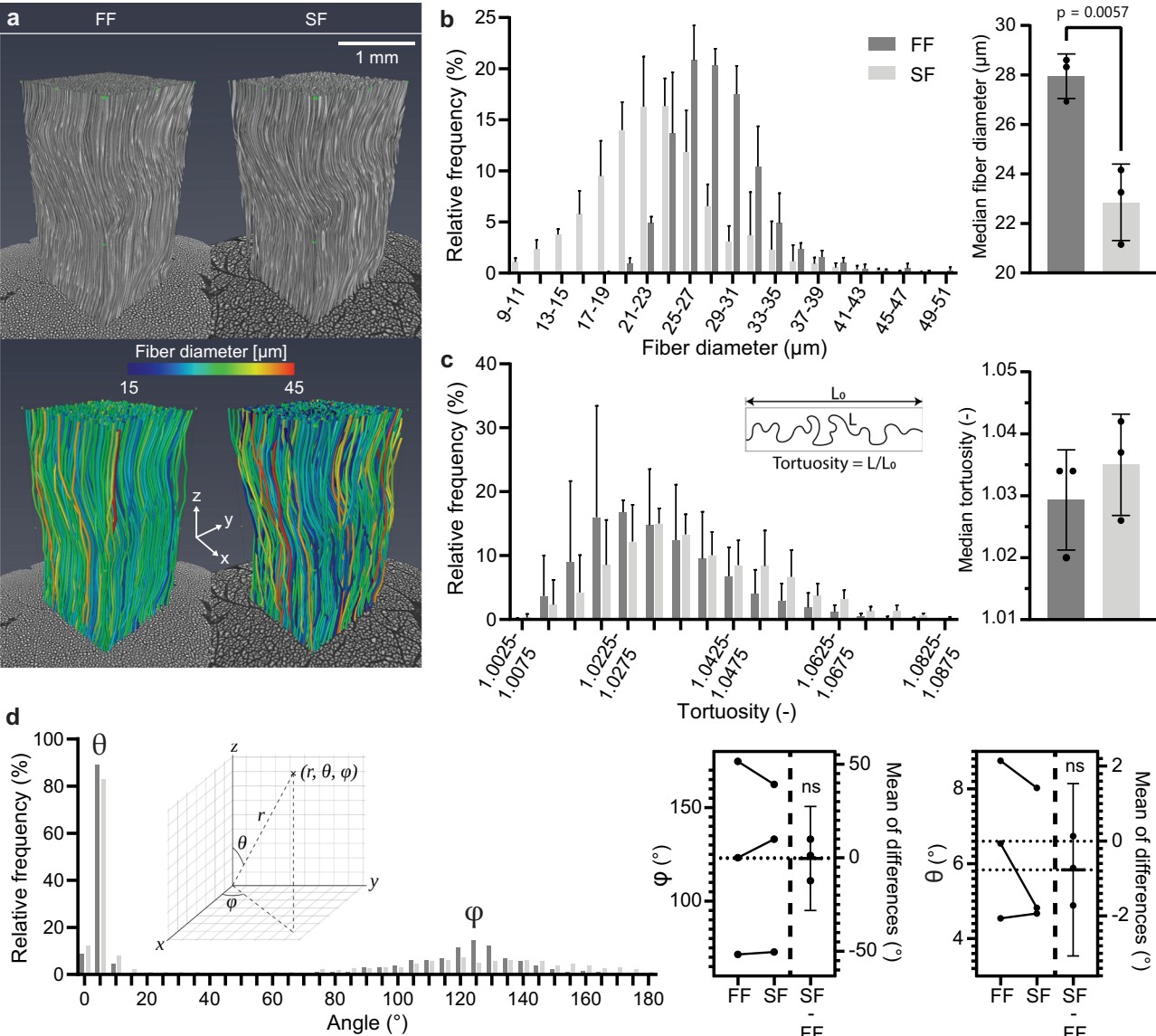

**Fig. 2 | Quantification of the effect of different freezing rates on the micro-structure of bovine muscle tissue. a** The fibrous microstructure of muscle tissue stained with Hf-WD POM and visualized with cryo-CECT both after fast freezing by submersion in isopentane at −78 °C (FF) and slow freezing in air at −80 °C (SF). On top, a volume rendering shows the VOI used for the structural analysis. Below, the fiber model is shown, in which the diameter of each fiber is indicated by the color scale. **b** Average histogram of the fiber diameter (left) and bar graph comparing the median fiber diameter (right) after FF and SF. **c** Average histogram of the fiber tortuosity (left) and bar graph comparing the median fiber tortuosity (right) after

FF and SF. **d** Average histogram of the orientation angles, θ and φ, after FF and SF (left). Estimation plots for the median φ and θ (right) showing the pair-wise com-parison of each sample after FF and SF, together with the mean of differences (SF-FF). The legend at the top of **b** applies to **b–d**. The bars in the histograms and bar graphs represent the mean, and the error bars indicate the standard deviation; $n = 3$ for each freezing rate, with >750 individual fibers measured in each VOI. Two-sided paired t-testing was conducted to compare groups (FF and SF) in the bar graphs of **b** and **c**. Significant p-values ($p < 0.05$) have been indicated in the bar graphs.

orientation of the fibers remained similar between sham-operated and TAC hearts (Fig. 5h). This quantitative analysis can scarcely be extracted from classical 2D histology and would require extensive manual work to stack and analyze the large number of 2D sections.

In addition to the fiber orientation, cryo-CECT allowed to quantify the diameter of the myocardium fibers in a VOI located in the septum, close to the base of the heart. From this fiber diameter quantification, a shift in relative frequency distribution towards larger fiber diameters was observed for TAC compared to sham-operated hearts. However, statistical significance between the sham and TAC group was not reached for the median fiber diameter (Fig. 5i). This analysis demon-strates the potential to obtain precisely spatially-defined 3D data from cryo-CECT, whereas results obtained from classical 2D histology would highly depend on the location and orientation of the 2D sections.

Fibrosis could only be observed in one sample of the TAC group (Fig. 5j–l). Interestingly, this heart was also the most hypertrophic among the TAC hearts, as quantified by the ratios of heart weight and volume to body weight (Fig. 5a, b). Interstitial and perivascular fibrosis could be identified on CECT and cryo-CECT images by darker gray regions inside the myocardium (Fig. 5j). The 3D visualization of these fibrotic regions, as validated by classical 2D histological sections using picrosirius red staining, allowed precise 3D localization of fibrosis within the myocardium. Four separate regions of severe fibrosis were identified, all located near the base and surrounding the left ventricle (Fig. 5k). Overlaying these regions with the fiber orientation map allowed the correlation between the location of the fibrosis relative to the distinct layers in the myocardial wall, which could not be obtained using classical 2D histology. The analysis showed that interstitial

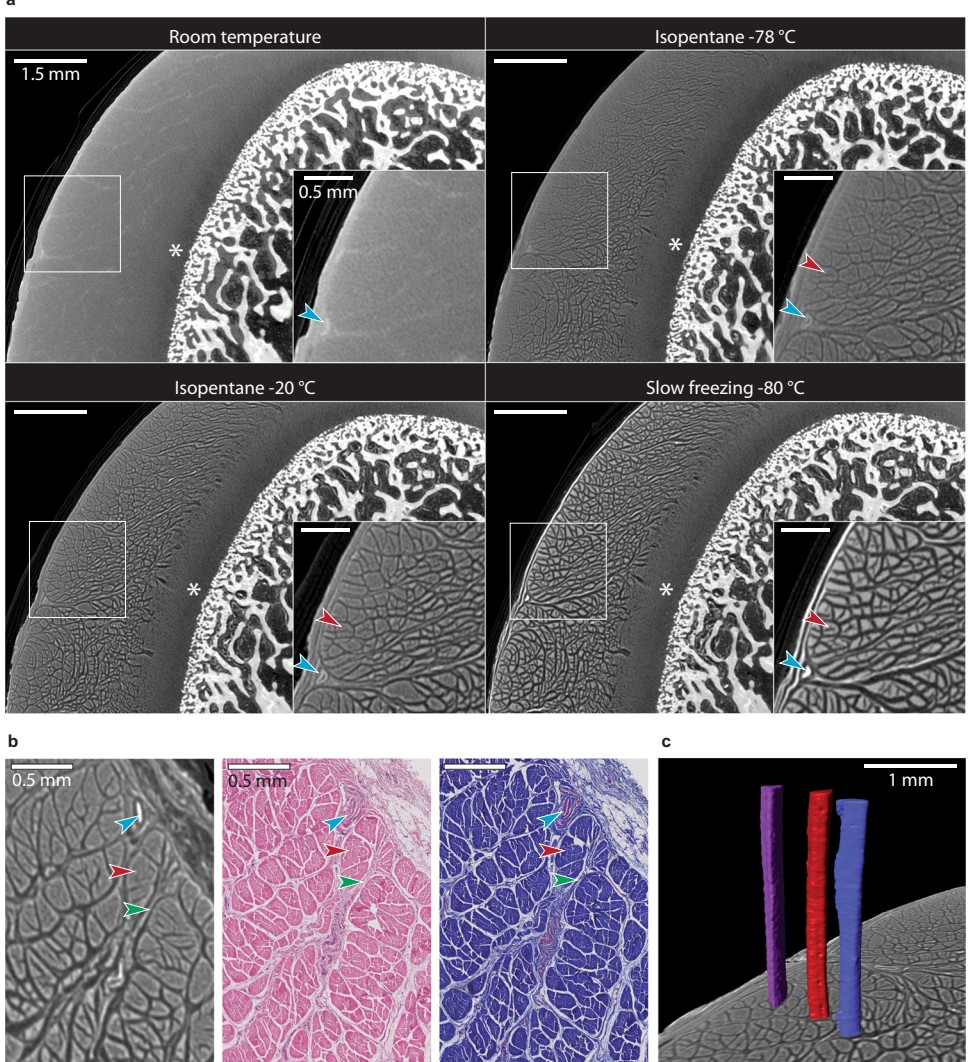

**Fig. 3 | The 3D visualization of individual collagen fibers in the tendon tissue at the porcine Achilles bone-tendon interface for different freezing rates.**
**a** Transverse microCT slices, perpendicular to the collagen fibers' direction, of the porcine Achilles bone-tendon ($n = 1$) stained with Hf-WD POM, obtained by conventional CECT (room temperature) or by cryo-CECT using different freezing methods: submersion in isopentane at −78 °C, submersion in isopentane at −20 °C or slow-freezing in air at −80 °C. A magnification (white square) is shown in the inset. The arrows indicate an individual collagen fiber in the tendon tissue (red arrow) and a blood vessel (blue arrow). The layer of non-mineralized fibrocartilage at the bone-tendon interface (*) is also indicated. **b** Comparison between a cryo-CECT image (isopentane −20 °C) and the matching histological slices stained with H&E (middle) and Masson's trichrome (right). A blood vessel (blue arrow), an individual tendon collagen fiber (red arrow) and the interfascicular matrix (green arrow) are indicated. **c** Volume rendering of a few collagen fibers in the tendon tissue showing the 3D arrangement of the fibers.

fibrosis occurred in both the layers of circumferential and vertical cardiac muscle fibers (Fig. 5l and Supplementary Video 2).

## Discussion

"Form follows function" reflects an established principle in biology[1]. The microstructure of biological tissues is closely linked to their in vivo functioning, and is often impacted by diseases causing malfunctioning. Hence, the 3D characterization of the tissue's microstructure in great detail is crucial. Unlike conventional 2D histological methods, X-ray based 3D histology -using CECT imaging- allows the 3D visualization of the complex and heterogeneous microstructure of soft biological tissues. However, current CECT protocols often induce tissue shrinkage and deformation, which could result in a misleading representation of the tissue's true microstructure and could also bias subsequent structural analyses. Moreover, certain tissue constituents, such as individual muscle or collagen fibers, remain challenging to visualize using conventional CECT. In this regard, we developed the technique

termed cryo-CECT, which offers a nondestructive approach that significantly expands the potential of CECT to visualize individual tissue constituents. Indeed, we demonstrated the ability of cryo-CECT to visualize the microstructure of skeletal and cardiac muscle tissue, as well as tendon tissue, down to the individual fiber level. Analogous to how Expansion Microscopy (ExM) broadened the abilities of conventional optical microscopy[56,57], cryo-CECT allowed to overcome the current limitations of CECT without the need for more advanced and less accessible X-ray based techniques such as synchrotron-based phase contrast CT.

We hypothesize that tissue constituents are visualized by cryo-CECT because of segregation between the formed ice and the CESA that is absorbed by the tissue (Fig. 6). Upon freezing, the water molecules contained in the tissue crystallize to form solid ice crystals. Since the CESA is insoluble in crystalline ice, segregation between the ice and the CESA is expected. Consequently, the formed ice crystals consist of water, whereas the CESA remains within the tissue's

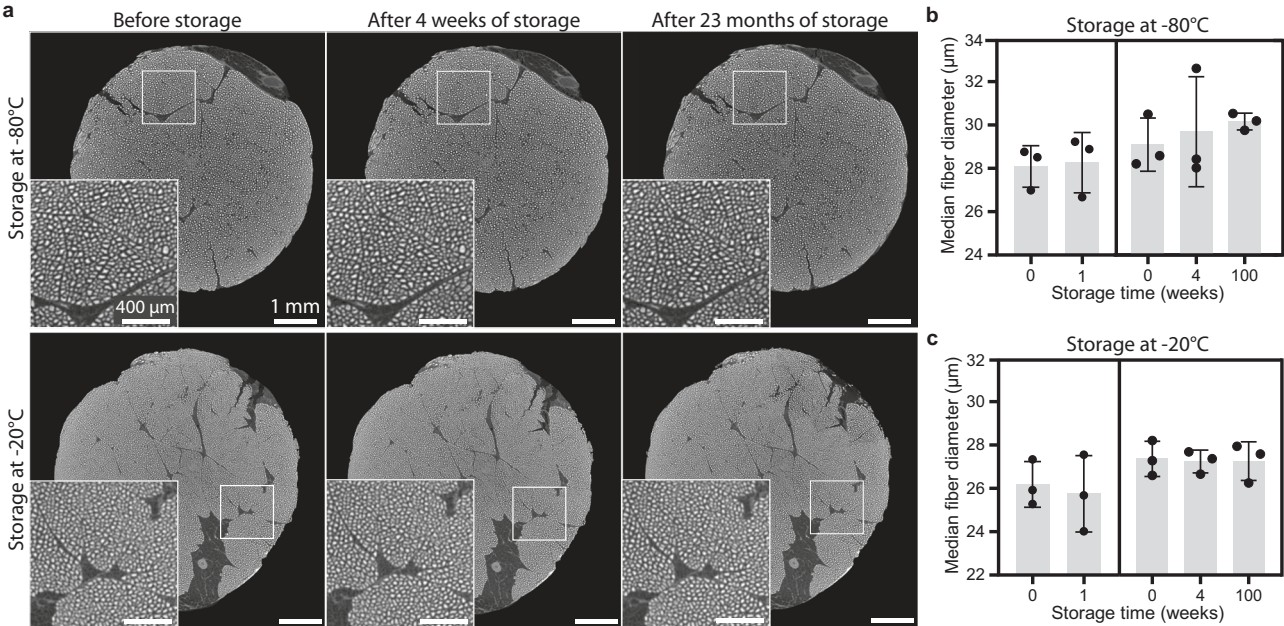

**Fig. 4 | Stability of the fast-frozen microstructure during low-temperature storage. a** Cryo-CECT images of the Hf-WD POM-stained muscle tissue immediately after fast freezing in isopentane at −78 °C and after storage for 4 weeks and for 23 months, at −80 °C (top) and −20 °C (bottom). Within each cryo-CECT image, a magnification (white square) is shown in the inset. Pair-wise comparison of the median fiber diameter after 1 week (left) or 4 weeks and 100 weeks (right) of storage at −80 °C (**b**) or storage at −20 °C (**c**). Bars represent the mean, and error bars indicate the standard deviation; $n = 3$ for each time point of storage with $n > 750$ individual muscle fibers measured in each VOI. Two-sided paired t-testing was conducted to compare the storage times of 0 and 1 week. For comparison of the three timepoints (0, 4 and 100 weeks), one-way analysis of variance with repeated measures, followed by a two-sided Tukey's test, was conducted. ns $p > 0.05$.

constituents. This explains the high contrast between the stained tissue and the surrounding ice. For cryo-CT, a similar process takes place with, instead of a CESA, the dissolved salts from the PBS buffer, which reveals a slight contrast between the unstained tissue containing the salts and the ice crystals. Ice formation can be considered as two subsequent processes: crystal nucleation and crystal growth[58]. At first, undercooling ($\Delta T = T_{melt} - T$) is necessary to surmount the energy barrier related to the phase change to form stable ice crystal nuclei. The amount of undercooling determines the critical size of the nuclei and, hence, the probability of stable nuclei formation. A high degree of undercooling, such as for fast freezing, results in numerous small nuclei. After crystal growth has ceased, small ice crystals are homogeneously distributed throughout the tissue, both intra- and extracellular. Contrarily, slow freezing is characterized by low degrees of undercooling. In this case, ice crystal nuclei will mainly form in the extracellular spaces, resulting in an unfrozen matrix with a high solute concentration outside the cells. The osmotic gradient in solute concentration facilitates the diffusion of intracellular water to the extracellular spaces, dehydrating the cells. The water will subsequently freeze in the extracellular spaces giving rise to large extracellular ice crystals and, in our case, a squeezed appearance of the fibers[58,59].

In this study, two different CESAs were evaluated for the visualization of skeletal muscle tissue using cryo-CECT (Fig. 1). Lugol's iodine was included because of its widespread use in conventional CECT imaging of muscle tissue. Nevertheless, this CESA is known to induce significant tissue shrinkage, which sacrifices the nondestructive character of CECT[20,21,31,52,53]. Therefore, Hf-WD POM was selected as a nondestructive alternative CESA, which prevents tissue shrinkage. In comparison to CECT, cryo-CECT enhances the fiber visualization considerably. Interestingly, this enhancement was more pronounced for Hf-WD POM staining, which could be explained by the sample dehydration caused by Lugol's iodine. Follow-up studies are planned to investigate the efficacy of other nondestructive CESAs such as CA4+[26,60] or different types of POMs[36] for cryo-CECT. In addition, a recent study by Dawood et al. reported that the use of a stronger phosphate buffer avoided the

acidification of the Lugol's iodine staining solution and, hence, almost completely prevented soft tissue shrinkage[61]. It would be interesting to evaluate the efficacy of this strongly buffered Lugol's iodine solution (B-Lugol) for cryo-CECT applications.

We demonstrated the added value of cryo-CECT imaging for the 3D imaging of tissue constituents. However, the range of potential applications for cryo-(CE)CT is not limited to the 3D histological characterization of biological tissues. The ability to visualize the microstructure of tissues solely by freezing the sample (cryo-CT), albeit with low image contrast (Fig. 1a, d), could be useful in applications where CESAs are precluded. For instance, cryo-CT could play an important role in the optimization of preservation strategies of biological tissues and organs, since there is still a lack of consensus concerning their preservation conditions[62,63]. In this regard, cryo-CT could provide valuable information about the ice formation and the tissue's integrity following different preservation methods. Furthermore, cryo-CT could potentially be applied in donor banks as routine quality control to evaluate both the presence of calcifications and the soft tissue microstructure of organs and tissues. The early detection of these reductions in donor material quality could prevent poor transplantation outcomes. Finally, cryo-(CE)CT could potentially also find an application in the food industry, in which freezing is a crucial process for the preservation of food tissues[64–70].

The contrast-enhancing mechanism of cryo-CECT implies that the cellular permeability, water content and fiber packing of a particular type of soft tissue will dictate the optimal freezing rate. Cryo-CECT imaging requires a certain degree of ice crystal formation and growth in order to visualize individual tissue constituents, such as collagen fibers. However, care should be taken to avoid excessive tissue deformation due to too slow freezing. Contrarily, excessively high freezing rates could lead to very small ice crystals or even vitrification of the water, which impedes the visualization of tissue constituents by cryo-CECT. In addition, high freezing rates increase the risk of inducing freezing cracks within the tissue. Based on the 3D microstructural analysis, we found that fast freezing with isopentane at −78 °C was optimal for

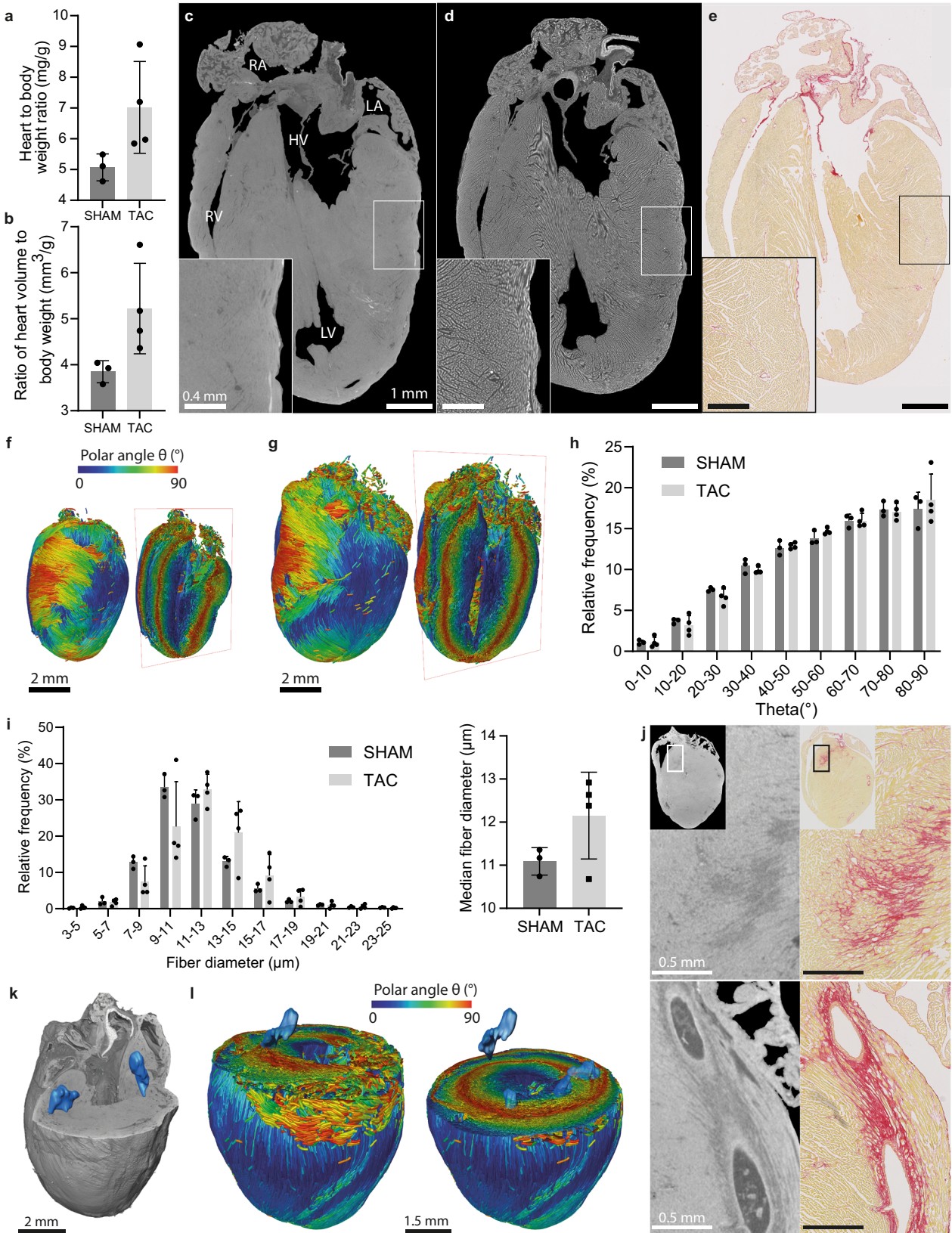

muscle tissue (Fig. 2), whereas collagen fibers in tendon tissue were optimally visualized after fast freezing with isopentane at −20 °C (Fig. 3). This demonstrates that, given an optimized freezing rate, diverse types of tissue constituents can be effectively visualized using cryo-CECT.

We showed the stability of the fast-frozen microstructure of the muscle tissue up to 23 months of storage both at −20 °C and −80 °C

(Fig. 4). The stability at −20 °C implies that the fast-frozen micro-structure also remains stable during long cryo-CECT scans that are acquired at −35 °C, since the recrystallization rate is known to increase with the ambient temperature. A possible reason for this stability is the glass temperature ($T_g$) of muscle tissue. If the storage temperature is lower than $T_g$, the unfrozen supersaturated phase will transform from a

**Fig. 5 | 3D histopathology of pressure overload-induced hypertrophic murine hearts following TAC surgery.** Bar graphs showing the mass (**a**) and volume (**b**) of the murine hearts of the sham and TAC group. Longitudinal sections of the same heart through the left and right ventricle, obtained by conventional CECT (**c**), cryo-CECT (**d**) and classical 2D histological sectioning followed by picrosirius red staining (**e**). RA right atrium, RV right ventricle, LA left atrium, LV left ventricle, HV heart valve. 3D spatial graph of the cardiac muscle fiber orientation of a sham heart (**f**) and a TAC heart (**g**). Fiber orientation is represented by the inclination, or polar angle θ and is indicated by the color scale. **h** Average histogram of the cardiac muscle fiber inclination comparing the sham and the TAC group. **i** Average histogram of the cardiac muscle fiber diagram (left) and a bar graph of the median fiber diameter (right) comparing the sham and the TAC group. **j** Sections obtained with CECT (left) and classical 2D histological sectioning followed by picrosirius red staining (right), showing the interstitial (top) and perivascular (bottom) fibrosis in a TAC heart. The rectangles in the inset indicate the location of the section. **k** Volume rendering of the CECT data of a TAC heart, showing the location of the regions that were severely affected by interstitial fibrosis (blue regions). **l** The regions of severe interstitial fibrosis (blue regions) overlaid with the fiber orientation map, clipped at two different heights. The fiber orientation is indicated as in **f**. Bars represent the mean, and error bars indicate the standard deviation; $n = 4$ for the TAC group and $n = 3$ for the SHAM group. To compare TAC and SHAM, two-sided unpaired t-testing was conducted on the data shown in the bar graphs of (**a, b, i**). *ns p > 0.05*.

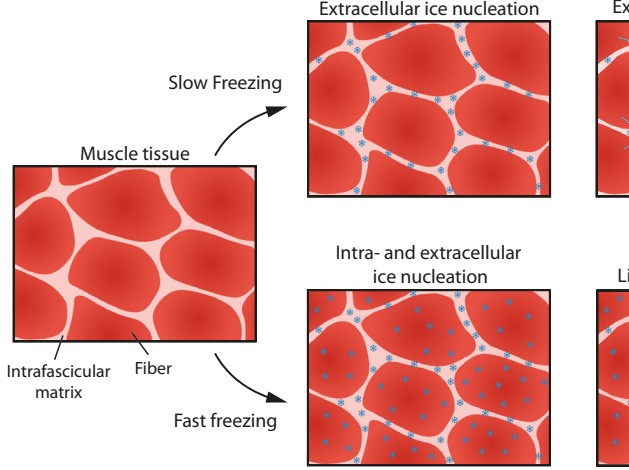
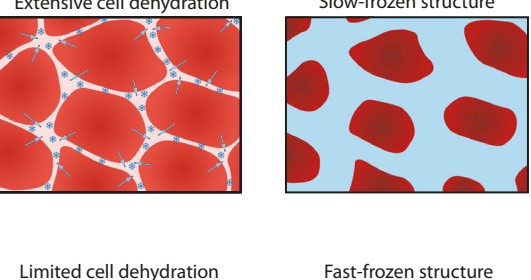

**Fig. 6 | Schematic overview of the ice formation during fast and slow freezing in muscle tissue.** Slow freezing of muscle tissue (top) leads to the formation of extracellular ice crystal nuclei (blue stars) outside of the muscle fibers (dark red). During the growth of these extracellular nuclei, the osmotic gradient leads to diffusion of intracellular water (light blue arrows) to the extracellular spaces (pink). As a result, the slow-frozen structure consists of compressed, dehydrated muscle fibers surrounded by large extracellular ice crystals. The high undercooling associated with fast freezing (bottom) leads to both intra- and extracellular ice crystal nuclei. The process of fast freezing also limits the amount of water diffusion resulting in a homogenous distribution of small ice crystals, both within and outside of the muscle fibers.

rubbery to a glassy state. This transition kinetically immobilizes the unfrozen matrix and greatly reduces the diffusion necessary for recrystallization processes. Therefore, $T_g$ is often regarded as the limiting temperature above which long-term deterioration is accelerated. Reported values for $T_g$ of bovine muscle tissue range from −60 °C to −5 °C[59]. However, more recent papers have consistently reported values around −13 °C[71,72]. Another important factor for ice recrystallization is temperature fluctuations during storage[73]. Hence, in this study, care was taken to use laboratory freezers that are designed to maintain a constant temperature.

The importance of histopathology in characterizing pathology-induced alterations in microstructure cannot be understated. However, laboratory tools for 3D microstructural analysis, such as fiber orientation and diameter assessment, are still scarcely available. In most studies assessing the heart, analyses are limited to in vivo hemodynamic properties and classical 2D histology. Watson et al. reported that diffusion tensor magnetic resonance imaging (DT-MRI) and ultrasound imaging were popular modalities to analyze the orientation of cardiac fiber bundles, while classical 2D histology remains the gold standard to visualize individual fibers[74]. For instance, a common practice to determine the cardiac muscle fiber diameter is to measure the cross-sectional area (CSA) of the fibers, based on classical 2D histological sections. However, the CSA highly depends on the sectioning orientation and location and, therefore, leads to a wide range of reported fiber diameters for wild-type mice (8 − 22 μm)[75–77]. In addition, the sample dehydration step required for classical 2D histological sectioning causes substantial tissue shrinkage, which can result in an erroneous quantification of the fiber diameter (Supplementary Fig. 3). This highlights the need for 3D imaging of individual fibers at high resolution for precise structural characterization. X-ray based techniques were reported to be efficient to visualize cardiac muscle fiber bundles and, in some cases, individual muscle fibers. However, they require invasive CESAs and/or dehydration of the samples[16]. To the best of our knowledge, cryo-CECT is the only lab-based microCT technique that allows to visualize single cardiac muscle fibers in full 3D without any dehydration step or the use of a shrinkage-inducing CESA. As opposed to classical 2D histology, the technique presented here allows the 3D analysis of the fibers in the entire heart and the quantification of their full 3D orientation and diameter. The ability to visualize this intricate organization of cardiac muscle fibers, responsible for torsion movement and an efficient ejection of the heart[78,79], is crucial to better understand the effects of pathologies altering these orientation patterns and the local arrangement of cardiac fibers, which remains poorly investigated in experimental models. Conventional CECT and cryo-CECT also enabled the 3D visualization and localization of TAC-induced fibrosis. Quantitative comparison of the severe fibrotic area fraction measured based on CECT and classical 2D histology resulted in similar values (0.83% and 1.11%, respectively) (Supplementary Fig. 4). This slight difference in area fraction is likely influenced by the imperfect image registration due to the sample deformation during sample handling and preparation for classical 2D histology. However, it is worth noting that, compared with CECT, classical 2D histology was able to visualize more finely dispersed regions of interstitial fibrosis thanks to its higher spatial resolution (Fig. 5j). The combination of the 3D quantitative analysis of the cardiac fibers with the localization of fibrotic tissue has the potential to reveal new insights in the correlations between fibrosis and a particular fiber orientation or a change in diameter.

Finally, our technique has some limitations. The use of an in-house developed, and thus non-commercially available, cryo-stage could be a constraint. However, although our cryo-stage offers valuable advantages compared to commercially available ones (homogeneous air cooling, temperature stability and no negative influence on the highest attainable spatial resolution), other cooling stages could also be used for cryo-CECT given that the stage (i) allows scanning at a sufficiently high spatial resolution and (ii) provides homogeneous and complete freezing of the sample. Furthermore, quantitative structural comparison between cryo-CECT and the gold standard (i.e., classical 2D histology) to determine the optimal freezing rate was not possible due to the substantial tissue shrinkage caused by the sample preparation for classical 2D histological sectioning (Supplementary Fig. 3). Another potential limitation of cryo-CECT is the risk of creating freezing cracks within the tissue, which is known to increase with higher freezing rates. Finally, the tissue-dependent optimal freezing rate requires a preliminary optimization study if new tissue types are to be investigated using cryo-CECT.

To conclude, cryo-CECT provides the means to visualize in 3D individual constituents of soft biological tissues, such as muscle and tendon fibers, that up to now either could not be revealed using conventional CECT or that could only be imaged using protocols inducing tissue shrinkage and/or dehydration. We demonstrated the relevance of cryo-CECT in the field of X-ray based 3D histo(patho)logy of soft biological tissues. Furthermore, cryo-CECT could also find its applications in the optimization of tissue preservation techniques, quality control of donor material and in the food industry. Besides muscle and tendon tissue, cryo-CECT could also unprecedentedly reveal the 3D microstructural organization of other soft biological tissue types, including heart valves, blood vessels, kidneys, brains and lungs.

## Methods

### Samples

Animal handling was approved by local authorities (Comité d'éthique facultaire pour l'expérimentation animale, 2021/UCL/MD/009, UCLouvain, Belgium) and performed in accordance with the Guide for the Care and Use of Laboratory Animals, published by the US National Institutes of Health[80]. Mice were housed with a 12 h/12 h light/dark cycle, with the dark cycle occurring from 6.00 p.m. to 6.00 a.m. Mice were observed daily with free access to water and standard chow.

Bovine psoas major muscle samples were supplied by a local farm (Jos Theys Boerderij, Belgium) where the samples were harvested from Hereford cows aged between 18 and 24 months. Care was taken to consistently select a cut at the same central position along the length of the muscle among different animals. The muscle was further dissected in smaller strips with a diameter of 5 mm and a length of 10 mm, with the muscle fibers orientated along the long axis. Following dissection, samples were fixed overnight using a 4% formaldehyde (FA) solution in PBS. Samples were then rinsed for 24 h in PBS and stored in fresh PBS at 4 °C until further analysis. During fixation and rinsing, samples were placed on a horizontal shaker plate at 4 °C.

The porcine Achilles bone-tendon insertion was a recuperation of experimental material, which was kindly donated by the IREC Experimental Surgery Laboratory (UCLouvain). The bone-tendon interface was cut in half along the sagittal plane using a rotary tool. Afterward, the sample was fixed for 48 h using a 4% FA solution in PBS. The sample was then rinsed for 48 h in PBS and stored in fresh PBS at 4 °C until further analysis. During fixation and rinsing, the sample was placed on a horizontal shaker plate at 4 °C.

Seven murine hearts were provided by the Pole of Cardiovascular Research (UCLouvain, IREC, CARD, Brussels), of which 4 were assigned to the transverse aortic constriction (TAC) group and 3 to the sham control group. Transverse aortic constriction (TAC) was performed on WT C57BL/6 J female mice (11-12 weeks of age), anesthetized using a single intraperitoneal injection of ketamine (100 mg/kg) and xylazine (5 mg/kg). A horizontal incision was made at the second intercostal space to expose the aortic arch. A 0-7 nylon ligature was tied between the innominate and left carotid arteries with an overlying 27-G needle, which was rapidly removed. The same procedure was performed in sham-operated mice, without the aortic constriction. 72 h later, echocardiographic analysis was performed to evaluate the aortic peak flow (Supplementary Table 1). Four weeks after surgery, mice were first anesthetized with a single intraperitoneal injection of anesthetic (ketamine 100 mg/Kg/xylazine 5 mg/Kg). The chest of the mice was opened to expose the heart. To completely remove the blood, a needle was inserted in the ventricles to perfuse the hearts with PBS. Then, hearts were excised and fixed with 10 ml of 4% paraformaldehyde for 48 h at 4 °C, followed by rinsing for 24 h in PBS. After fixation, the murine hearts were slightly dried and weighted with a scientific balance, and then immersed in PBS again. The volume of the hearts was measured by imaging the stained samples using CECT, followed by image segmentation.

### Contrast-enhancing staining agents

The Lugol's iodine ($I_2$KI) solution was prepared at a theoretically physiological osmolality (312 mOsm/kg) by dissolving 12.948 g KI and 6.474 g $I_2$ in 500 mL milli-Q water. This solution was then diluted 1:1 (volume ratio) with isotonic PBS (10 mM, pH = 7.4), thus preserving physiological osmolality, to obtain the 0.65% Lugol's iodine staining solution (consisting of 0.65 m/V% $I_2$ and 1.29% m/V% KI). The 1:2 hafnium(IV)-substituted Wells-Dawson polyoxometalate (Hf-WD POM; $K_{16}[Hf(\alpha_2-P_2W_{17}O_{61})_2] \cdot 19H_2O$) was synthesized as described in literature[81]. The Hf-WD POM staining solution was prepared by dissolving 35 mg/mL of Hf-WD POM in PBS. The osmolality of both staining solutions (Lugol's iodine: 303 mOsm/kg; Hf-WD POM: 302 mOsm/kg) was measured using a freezing point osmometer (3250 Single-Sample Osmometer, Advanced Instruments Inc., Norwood, MA). Based on the sequential microCT scans of the bovine muscle samples during the staining process (Supplementary Fig. 1), a staining time of 7 days, at which full staining was achieved, was selected for both CESAs. The porcine Achilles bone-tendon interface and the murine hearts were stained with Hf-WD POM for 14 days and 10 days, respectively. All samples were stained with a 25:1 solution-to-sample volume ratio, while placed on a horizontal shaker plate at room temperature.

### Freezing protocols

Prior to freezing, the samples were taken out of the staining solution and wrapped in 2 layers of Parafilm™ to avoid direct contact with the isopentane. For the slow freezing procedure, the sample was placed in a conventional laboratory freezer at −80 °C for at least 4 h. Fast freezing was performed by submerging the sample in cold isopentane (−78 °C) for 1 min. A bottle filled with isopentane was pre-cooled in the −80 °C freezer and placed inside a polystyrene box surrounded with dry ice to maintain the liquid's temperature at −78 °C. Additional freezing rates were explored by freezing the sample in isopentane at −20 °C or −160 °C. To achieve the former freezing protocol, the bottle of isopentane was placed in the −20 °C freezer overnight. For the latter freezing protocol, a metal beaker filled with isopentane was partially submerged in liquid nitrogen (−196 °C). As a result, the bottom half of the isopentane ($T_m$ = −160 °C) solidifies while the upper half remains liquid and reaches thermal equilibrium at −160 °C. The temperature of the isopentane was inspected using a low-temperature thermometer (Traceable™ LN2 Excursion-Trac™, Traceable™ Products, Webster, Texas, USA).

### High-resolution microfocus X-ray computed tomography imaging

MicroCT scans were acquired using a Phoenix NanoTom M (GE Measurement and Control Solutions, Germany) equipped with a 180 kV/15 W energy nanofocus X-ray tube. A diamond-coated tungsten target was used for all scans. For the cryo-(CE)CT scans,

the in-situ cryo-stage was installed. The cooling of the in-situ microCT cryo-stage is electrically driven by 6 (3 at each side) Peltier cooling modules (Ferrotec module 2020/324/060BS, Ferrotec Corporation, California, USA), which are liquid cooled by a 600 W external cooling system (Van der Heijden-Labortechnik GmbH, Germany). The temperature inside the cryo-stage is continuously measured by an NTC thermistor (10 kΩ) and monitored using LabVIEW (National Instruments Corp., Texas, USA).

A detailed overview of the acquisition parameters is provided in Table 1. The microCT datasets were reconstructed with the Datos|x software (GE Measurement and Control Solutions) and exported as XY slices (.tiff). An in-house developed MATLAB script was used to convert the 16-bit slices (.tiff) to 8-bit slices (.bmp), while simultaneously windowing the histogram range to the dynamic range of the dataset[82]. For the multi-scan of the murine hearts, 3 consecutive scans were acquired along the height of the sample (multi-scan) to image the entire volume. Afterward, zoom scans of the murine hearts were acquired near the base for the fiber diameter analysis.

## 3D structural fiber analysis

3D visualization and analysis of the datasets were performed using the Avizo software (Thermo Fisher Scientific, Bordeaux, France). First, the *Cylinder Correlation* module followed by the *Trace Correlation Lines* module was applied on either a volume of interest ($1.25 \times 1.25 \times 3$ mm$^3$) for the bovine muscle samples or on the entire volume for the murine hearts to identify and trace the individual muscle fibers. The used input parameters for both modules are provided in Supplementary Table 2. Prior to the fiber analysis of the murine hearts, the long axis of the interventricular septum was manually aligned with the vertical Z-axis to ensure a reproducible position. Tracing lines were filtered (TensorZZ > 0.85 and CurvedLength >2.5 mm for the bovine muscle and CurvedLength >0.2 mm for the murine hearts) to exclude any lines that were not associated with (cardiac) muscle fibers or that were at the edge of the VOI. The tortuosity and the orientation of the fibers, defined in spherical coordinates as the polar angle θ and the azimuthal angle φ, were calculated using the *Spatial Graph Statistics* module. Next, the *Fiber Shape Statistics* module was used to reconstruct the individual muscle fibers based on the trace lines using the watershed method. This fiber model was applied to calculate various morphometric parameters of the muscle fibers such as the curved length, the volume and the average diameter (Eq. 1).

$$Average\ diameter = 2 \sqrt{\frac{Volume}{\pi * Length}} \tag{1}$$

Different microCT datasets of the same sample were automatically registered and aligned using Avizo. DataViewer software (Bruker MicroCT, Kontich, Belgium) was applied to manually align the microCT slices to the histological sections.

## Classical 2D histological assessment

After microCT scanning, samples were rinsed in PBS for at least 2 days and embedded in paraffin. Sections of 5 μm thick were made using a microtome and stained with hematoxylin and eosin (H&E), Masson's trichrome or Sirius red for comparison with cryo-CECT. Histological sections were imaged using a SCN400 Slide Scanner (Leica Microsystems, Germany).

## Statistical analysis

GraphPad Prism 9 (GraphPad Software, California, USA) was used for the statistical analysis and data visualization. Structural parameters were calculated for each individual fiber inside the VOI and presented in histograms (relative frequency). The median of each parameter was calculated to quantify the central tendency of the data distribution. Two-sided paired t-testing was conducted to compare groups. For comparison of more than two groups, one-way analysis of variance

**Table 1 | MicroCT acquisition parameters**

| Experiment | Effect of CESA and freezing rate (Figs. 1 and 2) | Visualizing collagen fibers in tendon tissue (Fig. 3) | Ice recrystallization (Fig. 4) | Pressure overload-induced hypertrophic murine hearts (Fig. 5) | | | Volume changes during staining (Supplementary Fig. 1) |
|---|---|---|---|---|---|---|---|
| **Tissue type** | Bovine muscle | Porcine Achilles tendon insertion | Bovine muscle | Murine heart (overview) | Murine heart (multi-scan) | Murine heart (zoom) | Bovine muscle |
| Voxel size (μm) | 3.5 | 7 | 4.3 | 5.0 | 2.0–2.7 | 1.5 | 10 |
| Source voltage (kV) | 95 | 80 | 100 | 75 | 70–80 | 85 | 60 |
| Tube current (μA) | 155 | 310 | 170 | 250 | 112–163 | 75 | 650 |
| Exposure time (ms) | 500 | 500 | 500 | 500 | 500 | 1250 | 500 |
| Number of images | 2400 | 2100 | 2100 | 1800 | 2400 | 2400 | 1800 |
| Average[a] | 2 | 3 | 1 | 1 | 3 | 3 | 1 |
| Skip[a] | 1 | 1 | 1 | 0 | 1 | 1 | 0 |
| Scan time (min) | 60 | 70 | 18 | 15 | 3 × 86 | 206 | 15 |

[a]More information about these acquisition parameters can be found in the following refs. 83,84.

with repeated measures, followed by a two-sided Tukey's test, was conducted. *P* values below 0.05 were considered to be significant and are indicated in the bar graphs. The mean value of the different samples is indicated by the height of the bars. Individual data points for each sample are indicated in the bar graphs. Error bars represent the standard deviation.

## Reporting summary

Further information on research design is available in the Nature Research Reporting Summary linked to this article.

## Data availability

The microCT datasets generated and/or analyzed during the current study are not publicly available due to their considerable size, but are available from the corresponding author on request.

## Code availability

The in-house developed MATLAB script to convert the reconstructed 16-bit slices (.tiff) to 8-bit slices (.bmp or .jpg), while simultaneously windowing the histogram range to the dynamic range of the dataset is available at: https://github.com/contrast-team/histogram-windowing (DOI: 10.5281/zenodo.7034265)[82].

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

## Acknowledgements

We would like to thank Jos Theys (Jos Theys Boerderij) for the supply of bovine samples, Johan Vanulst and Joris Bleukx (KU Leuven) for the technical support in the design of the in-situ cryo-stage, Walid El Aazmani and Delia Hoffmann (UCLouvain) for the sectioning and colorimetric staining of the biological samples, and Markus Huber-Lang (University Hospital of Ulm), Nick van Gastel (UCLouvain) and Steve Stegen (KU Leuven) for reviewing the manuscript. The X-ray microCT images were generated at the KU Leuven XCT Core Facility (supported by Jeroen Soete, research expert). Ar.M. and G.K. acknowledge the internal support from KU Leuven (funding C24/17/052). Ar.M., G.K. and G.P. acknowledge the support from the SBO project of the Research Foundation Flanders (FWO; grant S007219N). C.P. and G.K. acknowledge the Action de Recherche Concertée (ARC 19/24-097)-Fédération Wallonie-Bruxelles and acknowledge the support from ASBL Jean Degroof-Marcel Van Massenhove funding and the UCLouvain Foundation Saint-Luc (RM2A project). Al.M. is supported by grant from the Fonds National de la Recherche Scientifique (FNRS), Belgium (T.0011.19; T.0009.21). This work is also supported by an ARC grant (ARC 18/23-094). L.B. acknowledges support by grants from FNRS and ARC. S.H. works as Senior Research Associate at FNRS, Belgium. T.B. acknowledges the French Community of Belgium in the framework of a FRIA grant (40004158). T.B., S.V. and W.D.B. acknowledge the support from an FWO project (grant G088218N). L.L. acknowledges the support from a project of the Fonds de Recherche Spécial (FSR) Jeunes Académiques and the UCLouvain (assistant position).

## Author contributions

Ar.M., G.K. and M.W. conceived and designed the study. Ar.M. and C.P. conducted most experiments including sample preparation and staining for microCT imaging, microCT acquisition and microCT data analysis. Al.M. contributed to the TAC surgery, the peak verlocity measurements and dissection of the murine hearts. T.B., S.V. and W.D.B. contributed to the development and synthesis of the Hf-WD POM and Lugol's iodine staining solutions. S.V. contributed to the microCT acquisition of the volume changes during staining experiment. L.L. contributed to the recrystallization experiment. G.P. contributed to the development of the fiber analysis workflow of the murine hearts in Avizo. L.B., C.B. and S.H. contributed to the interpretation of the TAC experiments. Ar.M., C.P. and G.K. analyzed and interpreted data. Ar.M. and G.K. wrote and revised most of the study, together with contributions of C.P. to the TAC section. G.K. and M.W. direct the study and supervised the work.

## Competing interests

The authors declare no competing interests.

## Additional information

**Correspondence and requests** for materials should be addressed to Greet Kerckhofs.

