## [Peer Review File · Nature Communications]

Cryogenic Contrast-Enhanced MicroCT Enables Nondestructive 3D Quantitative Histopathology of Soft Biological TissuesREVIEWER COMMENTS

Reviewer #1 (Remarks to the Author):

In this study, Maes et al. reported that data on the development of a new technic to qualitatively and quantitatively assess organ microstructure. The authors focused on muscle and collagen fibers imaging. After determining the best conditions for imaging, the authors provided an interesting analysis of murine hearts, comparing pressure-overload and sham conditions. They concluded that their cryo-CECT was a good approach to characterize the orientation and diameter of fibers in a 3D mode, as well as to quantitate fibrosis in the full organ. Multiple applications outside of the biological research area could also be interesting.

This is an interesting study providing a new way to assess (micro)structure and perform structure/function analysis. The main advantage is the opportunity to look at the 3D full organ. I have few comments for the authors regarding their manuscript.

1 – Tissue shrinkage/swelling was tested for 2 CECAs. The authors reported changes between control and CECA as well as for each CECA, differences from pre-staining stage to post staining (from 1 to 38 days). Did you perform statistical tests to compare both conditions and evolution over time?

2 – It is not clear how many samples have been imaged in all the experiments. In some, the n is presented in the legend of the figure, but for other one there is no information. For example, to define the best CECA (Fig1), did you perform only one scan for each condition? What is the reproducibility of image acquisition or to contrast the sample? Was it similar for both CECAs?

3 – Histology was used as gold standard to validate the best CT approach. On CT slice, you determine fibers diameter. Could you do the same analysis of histological slice to confirm the diameter observed in CT with the contrast agent was as close as possible as the diameter of the fibers measured in histology? The comparison of CT images between them has limitation, and the analysis and comparison of CT vs histology will strengthen the choice of the best agent and your conclusion regarding quantification of fiber structure by CT. This could also be applied to the interfascicular space.

4 – Could you provide data on the reproducibility of the post processing fiber segmentation?

5 – The authors tested the stability over time of their samples to assess microstructure. They tested it over 1 and 4 weeks. Could you discuss longer storage time and potential problem related to this? As perspective, you mention industrial this technology

6 – As discussed in a previous comment, could you quantitate fibrosis by CT and compared this to the quantification of the registered histological slice?

7 – Is it possible to perform a quantification of fiber diameter and fibrosis content in an entire heart? For animal-based experimental study, this could very useful?

8 – Discussion regarding other imaging modalities, with advantages and disadvantages of CT-based approach could be interesting.

Minor point

The TAC surgery did not “induce” aortic stenosis. This surgery models cardiac acute pressure-overload.

Reviewer #2 (Remarks to the Author):

The authors present a nifty & novel ex vivo imaging approach for 3D histology using cryo-preservation, staining agents and micro computed tomography.

The results are convincing and beautiful, and the argument is clearly made for a place for these techniques in the histological assessment of tissues.

Discussion around the advantages/disadvantages against traditional histology (beyond the 2D v 3D nature) would be valuable. Traditional histology allows for counterstaining, sub-nanometer microscopy imaging, immunohistochemistry, etc. Additionally, the effect of freezing on cell structure and fidelity would be of interest if higher resolutions were sought beyond fibre imaging.

Methodologically, the work is sound. I would encourage the authors to read the recent paper by Dawood et al (Sci Reports, 2021, <https://doi.org/10.1038/s41598-021-99202-2>) where buffering the pH of Lugol's is shown to drastically reduce tissue shrinkage rather than osmolality. This would potentially bring Lugol's back to the drawing board for further investigation.

The order of the protocol describing the murine heart scanning is confusing. Were animals killed and then anaesthetised? I assume not, but cannot make sense of this and would not be able to reproduce it in my lab as described.

In table 1, what does "skip" refer to?

Does sample orientation affect the 3D structural fibre analysis? If yes, how was reproducible position controlled?

The number of samples are not included consistently per sub-experiment. Also inclusion of n=? would be valuable in the figures/captions; particularly where graphs with error bars are provided and bar charts (addendum: and why not individual samples with error bar as is currently advised for data representation).

Overall, this is an excellent and interesting work.

Reviewer #3 (Remarks to the Author):

The manuscript entitled "Cryogenic Contrast-Enhanced MicroCT: a Nondestructive Approach to 1 3D Quantitative Histopathology of Soft Biological Tissues," by Maes et al describes a novel cyro-CECT imaging method and applies the technology to the characterization of heart tissue. Cryo-CECT provides a nondestructive 3D image of individual soft tissue constituents by imaging the stained sample in its frozen state. In doing so the tissue does not shrink or alter in size, thus overcoming a limitation of previous methods. Using cryo-CECT, the authors were able to nondestructively visualize 3D individual skeletal muscle and tendon fibers, which could not be achieved with conventional CECT. Moreover, the authors demonstrated the potential of the technique in the evaluation of murine myocardium and the effects caused by a transverse aortic constriction surgery.

Overall the paper is well written, will be of interest to a general readership, and the technique applicable to many areas of histo/pathology. The images of the fibers are excellent and very nice to see at this resolution. I recommend publication after the following comments are addressed.

Page 1.

microCT – is generally referred to as micro X-ray computed tomography not microfocused.

The two review papers on CEASs are from the authors laboratory – please include 2-3 references from other laboratories. A quick google search reveals these two papers (X-ray-Computed Tomography Contrast Agents - <https://doi.org/10.1021/cr200358s> and Evaluation of X-ray tomography contrast agents: A review of production, protocols, and biological applications --doi: 10.1002/jemt.23225. Epub 2019 Feb 20). Please do a web of science search

Page 2.

The authors state that "Combining the contrast-enhancement of this freezing step with prior sample staining (i.e. cryo-CECT) substantially enhanced the visualization of the muscle fibers." "It is worth highlighting that the cryo-stage was not used to freeze samples, but solely to keep the samples frozen during scanning." Do you stain and then cyro or cryo and then stain – the word combining is ambiguous. Please clarify the text. What happens if you cyro the sample and then image while still frozen?

Pages 6-7.

The authors discussion of cooling rates and storage conditions are useful and insightful for the method. The ability to store the sample at -20 C is a positive features compared to storing at -80 C. What happens to the tissue if it warms to RT. What does the image look like? Can you re-cryo the image and get the same results?

Pages 10-11.

The authors should also discuss this technique relative to expansion microscopy. Please see Expansion Microscopy and one paper by the Boden group: doi: 10.1126/science.1260088

Pages 10-11.

The authors do a good job describing the benefits of this technique. Please include a paragraph, before the conclusion, that states some of the limitations – which include a home build cryo-CT stage or other steps or requirements of the method.

REVIEWER COMMENTS

Reviewer #1 (Remarks to the Author):

In this study, Maes *et al.* reported that data on the development of a new technic to qualitatively and quantitatively assess organ microstructure. The authors focused on muscle and collagen fibers imaging. After determining the best conditions for imaging, the authors provided an interesting analysis of murine hearts, comparing pressure-overload and sham conditions. They concluded that their cryo-CECT was a good approach to characterize the orientation and diameter of fibers in a 3D mode, as well as to quantitate fibrosis in the full organ. Multiple applications outside of the biological research area could also be interesting.

This is an interesting study providing a new way to assess (micro)structure and perform structure/function analysis. The main advantage is the opportunity to look at the 3D full organ. I have few comments for the authors regarding their manuscript.

1 – Tissue shrinkage/swelling was tested for 2 CECAs. The authors reported changes between control and CECA as well as for each CECA, differences from pre-staining stage to post staining (from 1 to 38 days). Did you perform statistical tests to compare both conditions and evolution over time?

We thank the reviewer for this valuable suggestion as statistical tests were not performed for this particular experiment in the initial manuscript. We have now performed two statistical tests to evaluate (1) the effect of staining on the volume change for both CECAs compared to the PBS control and (2) the effect of staining time for each CECA individually.

The first test was performed with a mixed-effects model instead of a standard 2-way ANOVA since not all CECAs were tested on the same time points (“missing data points” in the analysis). Multiple comparisons were carried out using the Dunnett’s test with the PBS group as the control group. Pooling the difference in relative volume over time for each CECA resulted in significant differences compared to the PBS control group for both Hf-WD POM and Lugol’s iodine. The second test was done using one-way ANOVA, performed for each CECA individually, where we compared the relative volume change over time compared to the initial state. Multiple comparisons were carried out using the Dunnett’s test with the initial state as the control group. Interestingly, this showed a significant difference from the first day of staining with Lugol’s iodine, whereas Hf-WD POM staining only resulted in a significant difference from day 11 and onwards.

These statistical analyses have been added to Supplementary Figure 1, and the results have been explained further in the second paragraph of the first Results section (see comment AM7 in the manuscript).

2 – It is not clear how many samples have been imaged in all the experiments. In some, the n is presented in the legend of the figure, but for other one there is no information. For example, to define the best CECA (Fig1), did you perform only one scan for each condition? What is the reproducibility of image acquisition or to contrast the sample? Was it similar for both CECAs?

We thank the reviewer for pointing this out. To clarify, we have specified the n for Figure 1 (n=3), Figure 3 (n=1) and Figure 5 (n=3 for SHAM and n=4 for TAC).

To evaluate the CESAs (Fig. 1), we used 3 samples for each CESA. As shown in the figure below, imaging results were highly reproducible both in terms of CESA (Hf-WD POM & Lugol's iodine) and fast freezing. Regarding the reproducibility for Figure 1, it is also worth mentioning that the quantitative analysis which is presented in Figure 2 of the manuscript analyzed the same samples that were stained with Hf-WD POM and fast-frozen using isopentane at -78° C. The low variability in fiber diameter distribution within one condition (n=3) confirms the reproducibility.

Figure | The reproducibility of the freezing and the staining for individual samples. Cryo-CECT cross-sectional slices of the different samples within each CESA group showing the reproducibility of the imaging results for fast-frozen (isopentane -78°C) muscle tissue, stained with either Hf-WD POM or Lugol's iodine.

3 – Histology was used as gold standard to validate the best CT approach. On CT slice, you determine fibers diameter. Could you do the same analysis of histological slice to confirm the diameter observed in CT with the contrast agent was as close as possible as the diameter of the fibers measured in histology? The comparison of CT images between them has limitation, and the analysis and comparison of CT vs histology will strengthen the choice of the best agent and your conclusion regarding quantification of fiber structure by CT. This could also be applied to the interfascicular space.

During the sample preparation for 2D histological sectioning, and especially during the dehydration step, samples are known to undergo substantial shrinkage. This was confirmed by imaging a murine heart before and after paraffin embedding with the sample still in the paraffin block (Supplementary Fig. 3 & comment

AM14 in the manuscript). As this shrinkage has an important influence on the structural properties such as fiber diameter, spacing between fibers, etc., we have deliberately decided not to compare these structural parameters in a quantitative manner between cryo-CECT and the 2D histological sections. Instead, the comparison between cryo-CECT and classical 2D histology was used as a qualitative tool to validate the visualization of the various tissue constituents.

4 – Could you provide data on the reproducibility of the post processing fiber segmentation?

The post-processing fiber segmentation is performed semi-automatically in Avizo by a combination of the modules *Cylinder Correlation* and *Trace Correlation Lines*, in which the user needs to define input parameters that describe the cylindrical shape of the fibers, as well as how straight the fibers are. Once these are defined, the process runs automatically and is, therefore, also highly reproducible. To increase the reproducibility of our fiber analysis protocol, we have now included these input parameters in the Supplementary Materials (Supplementary table 2) and referred to this table in the materials and methods section (see comment AM19 in the manuscript).

5 – The authors tested the stability over time of their samples to assess microstructure. They tested it over 1 and 4 weeks. Could you discuss longer storage time and potential problem related to this? As perspective, you mention industrial this technology

We thank the reviewer for this interesting suggestion. To answer to this comment, we decided to image the 4-weeks samples again, both for -20°C and -80°C storage (n = 3 for each storage temperature), since these were still stored in the respective freezers. This way, we obtained an additional timepoint for 23 months of storage at both temperatures. The results have been added to Figure 4 of the manuscript (see comment AM9 in the manuscript). We observed no clear differences in the microstructure, both qualitatively and quantitatively, between the initial state and after 23 months of storage at either -80°C or -20°C.

6 – As discussed in a previous comment, could you quantitate fibrosis by CT and compared this to the quantification of the registered histological slice?

As mentioned in the response to the comment #3, we decided to not compare quantitative structural parameters between (cryo-)CECT and classical 2D histology due to the substantial shrinkage caused by the sample preparation for classical 2D histology (Supplementary Fig. 3). The goal of comparing (cryo-)CECT slices with the registered picrosirius red stained sections was to qualitatively validate the darker grey regions on the (cryo-)CECT images as regions of severe fibrosis.

7 – Is it possible to perform a quantification of fiber diameter and fibrosis content in an entire heart? For animal-based experimental study, this could very useful?

In this study, regions of severe fibrosis were identified and localized in the entire murine heart. In follow-up studies on myocardial fibrosis, quantification could easily be performed by calculating the volume/number/position of these fibrotic regions. Concerning the fiber diameter, we deemed the spatial resolution of the entire heart scans (voxel sizes ranging from 2 to 2.7 μm) not sufficient to accurately measure the diameter of the cardiac muscle fibers ($\sim 11 \mu\text{m}$). Therefore, we acquired zoom scans (voxel size of 1.5 μm) to locally measure the fiber diameter in a volume-of-interest (VOI) located in the heart's septum. It is worth mentioning that the local measurement of the fiber diameter in a VOI could be beneficial to identify small local differences in the individual myocardial layers, which could otherwise remain unnoticed if the entire heart is analyzed. In theory, it could be possible to image the entire heart at a sufficiently high spatial resolution. However, this would increase the number of individual scans (and amount of data; $\sim 30 \text{ GB}$ per scan) of the multi-scan, which need to be merged together afterwards to obtain a continuous reconstruction of the heart.

8 – Discussion regarding other imaging modalities, with advantages and disadvantages of CT-based approach could be interesting.

Apart from phase-contrast CT, which was already mentioned in the manuscript, we have included a discussion on the following imaging modalities in the introduction section: confocal microscopy, light sheet microscopy, optical coherence tomography and micro-MRI (see comments AM2 and AM4 in the manuscript). Several citations to (review) papers on the different imaging modalities have been added as well.

9 – Minor point

The TAC surgery did not “induce” aortic stenosis. This surgery models cardiac acute pressure-overload.

We thank the reviewer for pointing out this incorrect expression. Indeed, the TAC surgery does not induce a biological stenosis of the aorta. Instead, by constricting the transverse aorta, an artificial aortic stenosis is introduced. This in turn allows to model cardiac acute pressure overload. To avoid any confusion for the readers, we have left out this part of the sentence in the results section (see comment AM10 in the manuscript): “..., we applied cryo-CECT to murine hearts subjected to pressure overload following TAC surgery ~~inducing an aortic stenosis (Fig. 5).~~”

Reviewer #2 (Remarks to the Author):

The authors present a nifty & novel *ex vivo* imaging approach for 3D histology using cryo-preservation, staining agents and micro computed tomography.

The results are convincing and beautiful, and the argument is clearly made for a place for these techniques in the histological assessment of tissues.

1 - Discussion around the advantages/disadvantages against traditional histology (beyond the 2D v 3D nature) would be valuable. Traditional histology allows for counterstaining, sub-nanometer microscopy imaging, immunohistochemistry, etc. Additionally, the effect of freezing on cell structure and fidelity would be of interest if higher resolutions were sought beyond fibre imaging.

We thank the reviewer for this valuable suggestion. We have expanded the introduction on classical 2D histology accordingly (see comment AM1 in the manuscript): “Currently, the gold standard for *ex vivo* tissue imaging remains classical 2D histological assessment thanks to its high discriminative power, the wide range of available (counter)stains and the ability of performing immunohistochemistry or fluorescence microscopy. Prior to microscopy, the sections are generally stained to highlight, for example, specific cells or various constituents of the extracellular matrix (ECM)³. Despite its many advantages, classical 2D histology is inherently limited by its 2D nature and the single sectioning orientation⁴. The intricate 3D tissue microstructure is, therefore, only partially revealed using classical 2D histology. In addition, this technique can be prone to image artefacts, such as sample distortion, folds, cracks and shrinkage due to dehydration.”

In addition, the introduction now includes a discussion on more advanced optical imaging modalities, that allow 3D imaging to some extent: serial stacking of 2D slices, confocal microscopy, light sheet microscopy and optical coherence tomography.

2 - Methodologically, the work is sound. I would encourage the authors to read the recent paper by Dawood et al (Sci Reports, 2021, <https://doi.org/10.1038/s41598-021-99202-2>) where buffering the pH of Lugol's is shown to drastically reduce tissue shrinkage rather than osmolality. This would potentially bring Lugol's back to the drawing board for further investigation.

The results of the study by Dawood *et al.* are certainly very interesting and also confirm our observed link between the tissue shrinkage and the acidification of the solution when using PBS as a solvent. It would be valuable in future work to test the efficacy of the buffered Lugol's iodine solution (B-Lugol) for cryo-CECT applications. We have added the following text in the discussion (see comment AM13 in the manuscript):

“In addition, a recent study by Dawood et al. reported that the use of a stronger phosphate buffer avoided the acidification of the Lugol's iodine staining solution and, hence, almost completely prevented soft tissue shrinkage⁴⁸. It would be interesting to evaluate the efficacy of this strongly buffered Lugol's iodine solution (B-Lugol) for cryo-CECT applications.”

However, we should mention that we encountered some contradictory results when applying the B-Lugol solution to murine kidneys (Fig. below). Compared with a conventional Lugol's iodine solution in PBS (Lugol PBS), the B-Lugol solution indeed prevented the sample shrinkage almost completely. However, we did

observe a substantial change in staining behavior when changing the buffer solution from PBS to the stronger Sorensen's (phosphate) buffer. By normalizing the grey values between the two datasets (Lugol PBS and B-Lugol), based on two reference materials (Al_2O_3 and borosilicate), we are able to compare the level of X-ray attenuation, and thus the iodine content, of the kidney's structures based on the normalized grey values. Firstly, staining with the B-Lugol solution resulted in an overall reduction of the grey values of the kidney indicating that less iodine was taken up by the tissue, compared with Lugol PBS. Secondly, the specificity towards the adipose tissue located at the kidney's hilum was more pronounced for the B-Lugol than for Lugol PBS. This indicates that also the specificity of the staining is altered by changing the buffer solvent. These findings are in contradiction with the conclusion by Dawood *et al.*: "We showed that staining in Lugol's solution prepared in Sorensen's buffer (B-Lugol) leads to a stable pH and almost completely prevents soft-tissue shrinkage, without affecting the staining process or timing." Therefore, we believe that we should first investigate in more detail the effect of the buffer solution on the staining behavior of Lugol's iodine, as well as the chemical processes behind this process. We are currently in contact with the research group of Dawood *et al.* to analyze the reason behind these discrepancies in our results.

Figure | Changing the buffer solvent from PBS (Lugol PBS) to Sorensen's buffer (B-Lugol) alters the staining behavior of Lugol's iodine. a, The relative normalization of the grey values (GVs) between the datasets Lugol PBS and B-Lugol, based on the two reference materials (Al_2O_3 and borosilicate; BSi), with the Lugol PBS dataset used as reference. The average grey values of the two reference material beads, measured along the lines, are indicated underneath the beads. **b-c,** CECT cross-sectional image (left) and volume rendering (right) of the murine kidneys stained with Lugol PBS (**b**) or B-Lugol (**c**). Grey values are normalized between (b) and (c) and can, therefore, be used to directly compare iodine content in the different kidney structures between Lugol PBS and B-Lugol.

3 - The order of the protocol describing the murine heart scanning is confusing. Were animals killed and then anaesthetized? I assume not, but cannot make sense of this and would not be able to reproduce it in my lab as described.

We apologize for the misunderstanding. Indeed, it is not the case that the animals were first sacrificed and then anesthetized. We hope to have cleared up the protocol by reformulating it as follows (see comment AM16 in the manuscript):

“72 h later, echocardiographic analysis was performed to evaluate the aortic peak flow velocity (Supplementary Table 1). Four weeks after surgery, mice were first anesthetized with a single intraperitoneal injection of anesthetic (ketamine 100 mg/Kg / xylazine 5 mg/Kg). **The chest of the mice was opened to expose the heart. To completely remove the blood, a needle was inserted in the ventricles to perfuse the hearts with PBS. Then, hearts were excised and fixed with 10 ml of 4% paraformaldehyde for 48h at 4°C, followed by rinsing for 24h in PBS.**”

4 - In table 1, what does "skip" refer to?

“Skip” is an imaging parameter that can be chosen in the datos|x acquisition software of the XCT system (GE Nanotom M). For normal scans (not fast-scan mode), the GE Nanotom M uses a stepwise sample rotation, which means that the sample rotates over a small angle and then momentarily stops to allow image acquisition. However, this sudden change in rotation speed could lead to vibrations to the sample. The number of “skip” (0, 1, 2, ...) refers to the number of images that will be ignored (or skipped) after the sample has stopped rotating to allow sample stabilization. Further explanation on “skip” can be found in this publication: doi: [10.1093/gigascience/gix027](https://doi.org/10.1093/gigascience/gix027). In addition, skip is also an important parameter to minimize the negative effects of a potential latent image on the detector in-between two imaging projections. This is described in the user manual of the datos|x acquisition software as follows: “The number of skip images is an important value for a CT. These "blank images" are used to minimize the effects of the detector afterglow. Depending on the detector, sample and beam parameters, this can have a greater or lesser effect on the quality of the acquisition.” We have added these two references to Table 1 in the manuscript (see comment AM18 in the manuscript).

5 - Does sample orientation affect the 3D structural fibre analysis? If yes, how was reproducible position controlled?

During the 3D fiber analysis, the orientation parameters of the sample are measured relative to the global XYZ axes of the software. So, indeed, the sample orientation within the software affects the results by the orientation analysis. However, in our case, only the orientation of the hearts relative to the vertical Z-axis would influence the results since the orientation analysis only quantified the polar angle, or the inclination, of the fibers. To control a reproducible position, the long axis of the interventricular septum was manually aligned with the vertical Z-axis. We have altered the text in the results section as follows (comment AM11): “Using the polar angle θ , or inclination, of the cardiac muscle fibers in relation to the long axis of the interventricular septum, a distinction could be made between vertical fibers ($\theta \sim 0^\circ$), found at the subendocardial (inner) and subepicardial (outer) layers of the myocardium, and circumferential fibers ($\theta \sim$

90°), located in the mid-myocardium.” We have also added the following sentence in the materials and methods section (comment AM20): “Prior to the fiber analysis of the murine hearts, the long axis of the interventricular septum was manually aligned with the vertical Z-axis to ensure a reproducible position.”

6 - The number of samples are not included consistently per sub-experiment. Also inclusion of n=? would be valuable in the figures/captions; particularly where graphs with error bars are provided and bar charts (addendum: and why not individual samples with error bar as is currently advised for data representation).

We thank the reviewer for pointing this out. To clarify, we have specified the number of samples for Figure 1 (n=3), Figure 3 (n=1), Figure 5 (n=3 for SHAM and n=4 for TAC), Supplementary Figure 1 (n = 3) and Supplementary Figure 2 (n = 3). In addition, we have adapted the bar graphs by including the individual data points.

Overall, this is an excellent and interesting work.

We thank the reviewer for confirming the quality of our work.

Reviewer #3 (Remarks to the Author):

The manuscript entitled “Cryogenic Contrast-Enhanced MicroCT: a Nondestructive Approach to 3D Quantitative Histopathology of Soft Biological Tissues,” by Maes et al describes a novel cryo-CECT imaging method and applies the technology to the characterization of heart tissue. Cryo-CECT provides a nondestructive 3D image of individual soft tissue constituents by imaging the stained sample in its frozen state. In doing so the tissue does not shrink or alter in size, thus overcoming a limitation of previous methods. Using cryo-CECT, the authors were able to nondestructively visualize 3D individual skeletal muscle and tendon fibers, which could not be achieved with conventional CECT. Moreover, the authors demonstrated the potential of the technique in the evaluation of murine myocardium and the effects caused by a transverse aortic constriction surgery.

Overall the paper is well written, will be of interest to a general readership, and the technique applicable to many areas of histo/pathology. The images of the fibers are excellent and very nice to see at this resolution. I recommend publication after the following comments are addressed.

1 - Page 1. microCT – is generally referred to as micro X-ray computed tomography not microfocused.

Indeed, the terms “microfocus X-ray computed tomography” and “X-ray micro computed tomography” are currently both used in the literature and are abbreviated as “microCT”. The term “microfocus X-ray computed tomography” refers to the micrometer-scale focal spot size of the CT system, whereas “X-ray micro computed tomography” refers to the micrometer-scale spatial resolution at which the CT images are acquired.

The focal spot size and the spatial resolution are linked to each other: the size of the focal spot dictates the highest attainable spatial resolution. For instance, X-ray micro computed tomography could be performed both by microfocus X-ray computed tomography and nanofocus X-ray computed tomography. We prefer using the term “microfocus X-ray computed tomography” since it provides more information about the focal mode in which the CT system was applied. In addition, we prefer to use this term in order to remain consistent with our previous publications. However, the term “X-ray micro computed tomography” would also be correct in this study, as voxel sizes varied from 2 to 10 μm .

2 - The two review papers on CEASs are from the authors laboratory – please include 2-3 references from other laboratories. A quick google search reveals these two papers (X-ray-Computed Tomography Contrast Agents - <https://doi.org/10.1021/cr200358s> and Evaluation of X-ray tomography contrast agents: A review of production, protocols, and biological applications --doi: 10.1002/jemt.23225. Epub 2019 Feb 20). Please do a web of science search

We thank the reviewer to point out this lack of diversity in the review papers describing CECT and CESAs. The second suggested review paper (doi: 10.1002/jemt.23225) is indeed interesting as it covers the traditional *ex vivo* CESAs, *in vivo* medical perfusion contrast agents and nanoparticle-based staining agents. Hence, it has been added as a reference in the introduction. The first paper (doi: 10.1021/cr200358s), however, only covers *in vivo* perfusion contrast agents to be used in a clinical setting and is thus less

relevant in the scope of this study. In addition, we have cited two review papers on CECT in the introduction (see comments AM3 and AM5 in the manuscript):

- (doi: 10.1007/s11307-018-1246-3) X-ray-Based 3D Virtual Histology—Adding the Next Dimension to Histological Analysis, Albers *et al.*
- (doi: 10.1186/s12915-020-0753-2) X-ray computed tomography in life sciences, Rawson *et al.*

3 - Page 2. The authors state that “Combining the contrast-enhancement of this freezing step with prior sample staining (i.e. cryo-CECT) substantially enhanced the visualization of the muscle fibers.” “It is worth highlighting that the cryo-stage was not used to freeze samples, but solely to keep the samples frozen during scanning.” Do you stain and then cryo or cryo and then stain – the word combining is ambiguous. Please clarify the text. What happens if you cryo the sample and then image while still frozen?

In cryo-CECT, samples are first stained and then frozen, either by submersion in isopentane at various temperatures (-78°C, -20°C or -160°C) or by slow freezing in the -80°C freezer. We believed the order of staining and freezing would be clear by mentioning “with prior sample staining”. However, to avoid any ambiguity, we have replaced this sentence by “Performing this freezing step on samples that have been stained with Hf-WD POM or Lugol’s iodine (i.e. cryo-CECT) substantially enhanced the visualization of the muscle fibers.” (see comment AM8 in the manuscript).

After freezing the samples, they were kept frozen whilst being transferred to the cryo-stage. This way, we were able to preserve and image the frozen tissue microstructure, which depends on the freezing method. The in-house developed cryo-stage was designed to provide a homogeneously cooled air chamber to hold the sample, in contrast to most commercially available stages that rely on (inhomogeneous) contact cooling. In summary, using cryo-CECT, we image samples that have been stained beforehand, in the frozen state.

4 - Pages 6-7. The authors discussion of cooling rates and storage conditions are useful and insightful for the method. The ability to store the sample at -20 C is a positive feature compared to storing at -80 C. What happens to the tissue if it warms to RT. What does the image look like? Can you re-cryo the image and get the same results?

To evaluate the effect of thawing, we have imaged one of the Hf-WD POM-stained samples that was frozen by submersion in isopentane at -78°C and afterwards stored in the -80°C freezer (left image below). Upon thawing, the sample returned to its original state without any observable changes compared to other samples that were imaged at room temperature, prior to any freezing (e.g. Fig. 1 in the manuscript).

CECT image of a bovine muscle sample after thawing, that had been stained with Hf-WD POM and fast-frozen by submersion in isopentane at -78°C

The second question (“Can you re-cryo the image and get the same results?”) can be answered by a previous experiment, which was eventually not included in the manuscript. In this experiment, we evaluated whether slow freezing in air at -80°C resulted in a permanent alteration of the fibrous microstructure (right image below). To this end, samples were fast frozen in isopentane at -78°C and imaged before (FF1) and after slow freezing (FF2), as well as in the slow-frozen state (SF). A comparison of the tissue’s microstructure prior to and after the slow freeze cycle did not reveal distinct differences, both visually and quantitatively. This indicates that the tissue can undergo freeze-thaw cycles, even a slow-freezing cycle, without altering the imaging results.

Evaluating permanent alterations of the fibrous microstructure due to slow freezing. **a-c**, Cryo-CECT images showing the Hf-WD POM-stained muscle tissue's microstructure at different time points, as indicated by the experimental flow chart above. Within each image, a magnification (white square) is shown in the inset. First, the tissue was fast frozen and imaged to obtain an image of the initial microstructure (a; FF1). Next, the tissue was thawed, slow frozen and imaged again (b; SF). Finally, the tissue was thawed, fast frozen and scanned to obtain an image of the microstructure following the slow freezing step (c; FF2). **d**, Pair-wise comparison of the median fiber diameter at different time points FF1, SF and FF2. Bars represent the mean, and errors bars indicate the standard deviation; n=3 for each time point, n>750 individual muscle fibers measured in each VOL; *p < 0.05 and ns p > 0.05.

5 - Pages 10-11. The authors should also discuss this technique relative to expansion microscopy. Please see Expansion Microscopy and one paper by the Boden group: doi: 10.1126/science.1260088 μ

We thank the reviewer for bringing this interesting technique to our attention. By isotropically expanding the tissue/cell anchored to a hydrogel, Expansion Microscopy (ExM) allows to overcome the physical limitations of conventional optical microscopes (diffraction limited) and, thus, prevents the need for more advanced techniques and equipment. We believe there is a strong analogy between the role of ExM in the optical microscopy field and that of cryo-CECT in the microCT field, as cryo-CECT allows the visualization of certain tissue constituents that could not be visualized in a nondestructive manner using conventional CECT. In this regard, cryo-CECT overcomes the current limitations of CECT, and avoids the need for more advanced X-ray based techniques such as synchrotron-based phase contrast microCT.

We have added the following text in the discussion section (see comment AM12 in the manuscript) and cited the following 2 papers:

“Analogous to Expansion Microscopy (ExM) in the field of optical microscopy^{56,57}, cryo-CECT allowed to overcome the current limitations of CECT without the need for more advanced and less accessible X-ray based techniques such as synchrotron-based phase contrast CT.”

- (doi:10.1126/science.1260088) Expansion microscopy, Chen *et al.*
- (doi:10.1038/s41592-018-0219-4) Expansion microscopy: principles and uses in biological research, Wassie *et al.*

Despite this analogy between cryo-CECT and expansion microscopy, we believe the two techniques are too dissimilar on a technical level to dedicate a separate paragraph in the discussion section on this comparison. First of all, expansion microscopy is based on the physical expansion (4.5x linear, or 100x in volume) of the cells/tissue using a hydrogel, whereas, with cryo-CECT, we try to minimize any volume changes or tissue deformations by optimizing both the CESA and the freezing rate. Secondly, the two techniques are based on different imaging modalities. Expansion microscopy is an optical microscopy technique and is often combined with fluorescence microscopy, whereas cryo-CECT is an X-ray based imaging modality. Thirdly, the two techniques are used to investigate different length scales. The spatial resolution of expansion microscopy can reach tens of nanometers, which allows the visualization of the cell's organelles such as microtubuli. In contrast, cryo-CECT is limited to a spatial resolution of a few microns. However, the larger field-of-view of cryo-CECT allows the 3D imaging of entire organs, whereas expansion microscopy is limited to the imaging of thick slices of about 100 μm .

6 - Pages 10-11. The authors do a good job describing the benefits of this technique. Please include a paragraph, before the conclusion, that states some of the limitations – which include a home build cryo-CT stage or other steps or requirements of the method.

We thank the reviewer for this suggestion. Indeed, a paragraph on the limitations of cryo-CECT was still lacking in the discussion section. We have added the following paragraph in the discussion section just before the conclusion (see comment AM15 in the manuscript):

“Finally, our technique has some limitations. The use of an in-house developed, and thus non-commercially available, cryo-stage could be a constraint. However, although our cryo-stage offers valuable advantages compared to commercially available ones (homogeneous air cooling, temperature stability and no negative influence on the highest attainable spatial resolution), other cooling stages could also be used for cryo-CECT given that the stage (i) allows scanning at a sufficiently high spatial resolution and (ii) provides homogeneous and complete freezing of the sample. Another potential limitation of cryo-CECT is the risk of creating freezing cracks within the tissue, which is known to increase with higher freezing rates. Finally, the tissue-dependent optimal freezing rate requires a preliminary optimization study if new tissue types are to be investigated using cryo-CECT.”

REVIEWER COMMENTS

Reviewer #1 (Remarks to the Author):

I read with interest the revised version of the manuscript by Maes et al. The authors did a great job and adequately addressed almost all the comments raised at the time of the first review.

I still have two minor comments.

1 – I understand that histology cannot be used as gold standard based on the important shrinkage of the samples related to this technic. However, the lack of gold standard approach to comparison of CT images and determine the best CT protocol to characterize the sample should be acknowledge in the limitation section.

2 – Regarding the quantification of the fibrosis by CT and registered histology, the relative proportion of fibrosis on a specific section of the sample should be concordant even if shrinkage of tissue was observed in one approach compared to the other. I would expect same percentage of fibrosis in both images, with or without shrinkage. Or, do you expect differences in shrinkage according to nature of tissue (i.e. fibrosis vs myocardial fiber)? This should be addressed in the manuscript (data presented or discussion/limitations).

Reviewer #2 (Remarks to the Author):

My comments have been adequately addressed.

Reviewer #3 (Remarks to the Author):

The authors have addressed both my major and minor concerns. They have add the additional data and references. The manuscript is improved and I recommend publication.

Reviewer #1 (Remarks to the Author):

I read with interest the revised version of the manuscript by Maes et al. The authors did a great job and adequately addressed almost all the comments raised at the time of the first review.

We once again thank the reviewer for his/her valuable remarks, improving the manuscript.

I still have two minor comments.

1 – I understand that histology cannot be used as gold standard based on the important shrinkage of the samples related to this technic. However, the lack of gold standard approach to comparison of CT images and determine the best CT protocol to characterize the sample should be acknowledge in the limitation section.

We agree with the reviewer that this should be stated as a limitation. The missing quantitative comparison with the gold standard has now been addressed in the limitations section as followed:

“Furthermore, quantitative structural comparison between cryo-CECT and the gold standard (i.e., classical 2D histology) to determine the optimal freezing rate was not possible due to the substantial tissue shrinkage caused by the sample preparation for classical 2D histological sectioning (Supplementary Fig. 3).”

2 – Regarding the quantification of the fibrosis by CT and registered histology, the relative proportion of fibrosis on a specific section of the sample should be concordant even if shrinkage of tissue was observed in one approach compared to the other. I would expect same percentage of fibrosis in both images, with or without shrinkage. Or, do you expect differences in shrinkage according to nature of tissue (i.e. fibrosis vs myocardial fiber)? This should be addressed in the manuscript (data presented or discussion/limitations).

We thank the reviewer for this interesting remark. Indeed, assuming that the degree of shrinkage is similar for myocardial and fibrotic tissue, the area fraction of the fibrosis to the entire tissue could be compared between CECT and classical 2D histology. We added these results as a supplementary figure (Supplementary Fig. 4) and discussed them in the discussion section:

“Quantitative comparison of the severe fibrotic area fraction measured based on CECT and classical 2D histology resulted in similar values (0.83% and 1.11%, respectively) (Supplementary Fig. 4). This slight difference in area fraction is likely influenced by the imperfect image registration due to the sample deformation during sample handling and preparation for classical 2D histology. However, it is worth noting that, compared with CECT, classical 2D histology was able to visualize more finely dispersed regions of interstitial fibrosis thanks to its higher spatial resolution (Fig. 5j).”

Reviewer #2 (Remarks to the Author):

My comments have been adequately addressed.

We once again thank the reviewer for his/her valuable remarks, improving the manuscript.

Reviewer #3 (Remarks to the Author):

The authors have addressed both my major and minor concerns. They have add the additional data and references. The manuscript is improved and I recommend publication.

We once again thank the reviewer for his/her valuable remarks, improving the manuscript.